# P4-ATPases control phosphoinositide membrane asymmetry and neomycin resistance

Bhawik K. Jain [1,3] ✉, H. Diessel Duan [2,3], Christina Valentine[1], Ariana Samiha[1], Huilin Li [2] & Todd R. Graham [1] ✉

The aminoglycoside antibiotic neomycin has robust antibacterial properties, yet its clinical utility is curtailed by its nephrotoxicity and ototoxicity. The mechanism by which the polycationic neomycin enters specific eukaryotic cell types remains poorly understood. In budding yeast, *NEO1* is required for neomycin resistance and encodes a phospholipid flippase that establishes membrane asymmetry. Here we show that mutations altering Neo1 substrate recognition cause neomycin hypersensitivity by exposing phosphatidylinositol-4-phosphate (PI4P) in the plasma membrane extracellular leaflet. Cryogenic electron microscopy reveals PI4P binding to Neo1 within the substrate translocation pathway. PI4P enters the lumen of the endoplasmic reticulum and is flipped by Neo1 at the Golgi to prevent PI4P secretion to the cell surface. Deficiency of the orthologous ATP9A in human cells also causes exposure of PI4P and neomycin sensitivity. These findings unveil conserved mechanisms of aminoglycoside sensitivity and phosphoinositide homoeostasis, with important implications for signalling by extracellular phosphoinositides.

Neomycin was first discovered in the 1940s and has been widely used for decades to treat bacterial infections[1,2]. It is derived from *Streptomyces fradiae*, a species of soil bacteria, and is known for its broad-spectrum antibacterial activity, particularly against Gram-negative bacteria[1]. Neomycin exerts its antibacterial effects by binding to bacterial ribosomes and inhibiting protein synthesis, and it is also capable of disrupting translation in eukaryotic cells[3,4]. However, most mammalian cell types are resistant to neomycin because the polycationic antibiotic cannot penetrate into the cell[2,5]. Despite this general resistance, aminoglycosides such as neomycin can cause various toxic effects on mammalian tissues. These include convulsions following intracisternal injection and inhibition of neuromuscular and ganglionic transmission, as well as nephrotoxicity and ototoxicity affecting the cochlea and vestibular system[6]. Toxicity of neomycin is attributed to its ability to enter particular cell types, such as the hair cells of the inner ear[7–9]. The precise biochemical mechanisms behind the neomycin sensitivity of specific eukaryotic cells remain unclear.

Neomycin specifically binds to phosphatidylinositol-4-phosphate (PI4P) and phosphatidylinositol 4,5-bisphosphate (PI(4,5)P$_2$)[8,10–12], lipids that play crucial roles in signal transduction, membrane trafficking and cytoskeletal dynamics through their ability to recruit and activate various effectors[13]. Phosphatidylinositol kinases synthesize phosphoinositides (PIPs) in the cytosolic leaflet of the Golgi, plasma membrane and endolysosomal system[14]. Most other phospholipids in the cell are produced in the cytosolic leaflet of the endoplasmic reticulum (ER). An ER scramblase allows rapid and energy-independent flip-flop, so newly synthesized phospholipids will populate both membrane leaflets[15–18]. As the membrane flows through the Golgi to the plasma membrane, energy-dependent phospholipid flippases in the P4-ATPase family transport phosphatidylserine (PS) and phosphatidylethanolamine (PE) from the lumenal or extracellular leaflet to the cytosolic leaflet, while leaving phosphatidylcholine and most sphingolipids behind[19]. Thus, the plasma membrane is asymmetric with

---

[1]Department of Biological Sciences, Vanderbilt University, Nashville, TN, USA. [2]Department of Structural Biology, Van Andel Institute, Grand Rapids, MI, USA. [3]These authors contributed equally: Bhawik K. Jain, H. Diessel Duan. ✉e-mail: bhawik.kumar.k.jain@vanderbilt.edu; tr.graham@vanderbilt.edu

PS/PE and PIPs enriched in the cytosolic leaflet and phosphatidylcholine and sphingolipids populating the extracellular leaflet.

P4-ATPases are highly conserved and budding yeast *Saccharomyces cerevisiae* expresses five P4-ATPases: Neo1, Drs2, Dnf1, Dnf2 and Dnf3. These lipid flippases localize to different cellular compartments and transport different lipid substrates[20–23]. Neo1 and Drs2 are Golgi-localized PS/PE flippases, orthologous to human ATP9A/B and ATP8A1/2, respectively, required to establish plasma membrane PS/PE asymmetry[21,24–26]. PIPs were not considered P4-ATPase substrates because they are not synthesized at the ER and presumably would not have access to the lumenal leaflet of the secretory pathway. However, PI4P can be delivered to the ER by lipid transfer proteins in the oxysterol-binding protein family, where it is exchanged for PS or cholesterol/ergosterol[27–30], raising the possibility that PI4P can flip-flop to the ER lumen and enter the secretory pathway.

This study reveals that PI4P enters the secretory pathway's lumen, is transported to the Golgi by COPII-dependent vesicular transport and is flipped by Neo1 back to the cytosolic leaflet. Abrogation of the ability of Neo1 to bind PI4P allows transport of this lipid to the extracellular leaflet of the plasma membrane, where it binds neomycin. We provide structural and biochemical evidence that PI4P is a transport substrate of Neo1 and evidence that this pathway is conserved in human cells.

## Results

### Neo1 transport pathway mutations cause neomycin sensitivity

*NEO1* was originally discovered in a screen for overexpressed genes that conferred neomycin resistance to wild-type (WT) *S. cerevisiae*[31]. Disruption of *NEO1* is lethal, but temperature-conditional mutants have been generated (*neo1-1*), and these cells are extremely sensitive to neomycin at the permissive growth temperature[32]. In addition, we previously isolated separation-of-function mutations in the Neo1 substrate translocation pathway (Fig. 1a, boxed region) that cause aberrant exposure of only PS, only PE or both substrates in the extracellular leaflet of the plasma membrane[24] (Fig. 1a,b). To ascertain if neomycin sensitivity is due to loss of PS or PE asymmetry, we examined the growth of these *neo1* mutants on solid media with or without neomycin (Extended Data Fig. 1). A subset of the mutants including S221L, R247L, S452Q and P456A were sensitive to neomycin and displayed an intermediate growth phenotype relative to WT cells and the *neo1-1* mutant (Fig. 1c), which exposes PS and PE[19]. We also observed growth inhibition of these neomycin-sensitive (neo1ˢ) mutants in liquid media containing as little as 50–100 μg ml⁻¹ neomycin, whereas WT cells are resistant to more than 1,000 μg ml⁻¹ neomycin (Fig. 1d). Curiously, the pattern of neomycin sensitivity did not correlate well to loss of PS or PE asymmetry. For example, the Q209G and S452Q Neo1 mutants both expose PS, but Q209G is neomycin resistant, while S452Q is neomycin sensitive[24] (Fig. 1b–d). Similarly, Q193A exposes PE but is resistant to neomycin, while the PE-exposing S221L and R247L mutants are neomycin sensitive. To further address the relationship between PS/PE asymmetry and neomycin sensitivity, we examined other flippase mutants (*drs2Δ*, *dnf1,2Δ* and *dnf3Δ*) that exhibit a loss of PS/PE membrane asymmetry and found that all are neomycin resistant (Fig. 1e). Dop1 and Mon2 are large cytosolic proteins that interact with Neo1 and are required for Neo1 activity[33,34]. Consistently, *dop1-1* and *mon2Δ* mutants exhibit the same degree of neomycin sensitivity as *neo1-1* (Fig. 1e,f). Consequently, neomycin sensitivity does not appear to result from loss of PS/PE asymmetry and is a unique phenotype of *neo1* relative to the other P4-ATPase mutants. Therefore, we hypothesized that Neo1 transports an unknown lipid substrate linked to neomycin sensitivity.

### Neomycin-sensitive mutants expose PI4P

Neomycin binds tightly to PIPs, but the biological importance of this interaction is unclear[8,10–12]. Budding yeast express a phosphatidylinositol 4-kinase (PI4-kinase, Stt4) and a phosphatidylinositol 4-phosphate 5-kinase (PI4P-5-kinase, Mss4) to produce PI4P and (PI(4,5)P₂), respectively, in the cytosolic leaflet of the plasma membrane[35–37]

(Fig. 2a). To test whether *neo1* mutants lose asymmetry of PI4P and/or PI(4,5)P₂ and expose PIPs in the extracellular leaflet, we probed cells with recombinantly purified green fluorescent protein (GFP)-tagged P4C domain (PI4P binding domain from SidC), specifically binding to PI4P, and the PLC$_{PH}$, pleckstrin homology (PH) domain of phospholipase C delta, which binds to PI(4,5)P₂ (refs. 38–40) (Extended Data Fig. 2g). Intriguingly, at the non-permissive growth temperature (38 °C), only GFP–SidC$_{P4C}$ bound to *neo1-1* spheroplasts (cells lacking the cell wall), with GFP–PLC$_{PH}$ failing to bind. Notably, the GFP–SidC$_{P4C}$ fluorescence predominantly localized to the cell periphery, exhibiting no discernible intracellular fluorescence (Fig. 2b). This distinct pattern of GFP–SidC$_{P4C}$ binding to the cell surface was observed in all other neomycin-sensitive Neo1 mutants as well. By contrast, neither probe bound to the neomycin-resistant, PS-exposing Q209G or PE-exposing Q193A mutants (Fig. 2c and Extended Data Fig. 2e).

To validate the specificity of GFP–SidC$_{P4C}$ for PI4P, we used a GFP–SidC$_{P4C}$(R652Q) variant with impaired PI4P recognition[38], which failed to bind to *neo1-1* cells (Fig. 2d and Extended Data Fig. 2f). We costained Neo1-S221L cells with GFP–SidC$_{P4C}$ and propidium iodide to distinguish between live and dead cells. Most of the cells in the population were stained with GFP–SidC$_{P4C}$, with very few instances of GFP–SidC$_{P4C}$-positive cells costaining with propidium iodide (Extended Data Fig. 2c,d). Thus, cells exposing PI4P were alive and capable of maintaining the permeability barrier to propidium iodide. Importantly, *dop1-1* and *mon2Δ* cells but none of the other flippase mutants exposed PI4P (Extended Data Fig. 2a,b). Thus, a strong correlation exists between mutants that expose PI4P in the outer leaflet and neomycin sensitivity. These findings also suggest that Neo1 is a PI4P flippase that normally prevents exposure of PI4P in the extracellular leaflet.

### Extracellular PI4P is produced by the PI4-kinase Stt4

PI4P is synthesized in the cytosolic leaflets of the plasma membrane and late Golgi by the PI4-kinase Stt4 and Pik1, respectively[36,41]. To explore the source of extracellular PI4P in *neo1* mutants, we created *neo1 stt4-4* and *neo1 pik1-83* double mutants. We assayed these double mutants for neomycin sensitivity at a semipermissive growth temperature where the *stt4-4* (33 °C) and *pik1-83* (34 °C) alleles would be partially defective[36,37,42,43] (Extended Data Fig. 3a). *STT4* and *PIK1* are essential genes, so complete inactivation of these temperature-sensitive alleles would prevent growth. Attenuating PI4P synthesis with *stt4-4* suppressed neomycin sensitivity caused by *neo1-S221L* or *neo1-S452Q*, as indicated by better growth of the *neo1 stt4-4* double mutants relative to the *neo1* single mutant in the presence of neomycin (Fig. 2e). By contrast, the *pik1* mutation had no discernible effect on neomycin sensitivity of *neo1* mutants (Fig. 2f). Moreover, a PI4P-5-kinase mutation (*mss4-102* (ref. 37)) did not significantly influence neomycin sensitivity in the *neo1* background (Extended Data Fig. 3b,c). Collectively, these results indicate that *neo1* sensitivity to neomycin is primarily attributed to PI4P synthesized by Stt4 at the plasma membrane. No loss of PI(4,5)P₂ asymmetry was detected in the *neo1* mutants nor did we detect a significant contribution of this lipid to neomycin sensitivity of yeast (Extended Data Fig. 3d).

### ATP9A depletion causes PI4P exposure in mammalian cells

ATP9A and ATP9B are the mammalian orthologues of Neo1 (ref. 44), and sequence comparison indicates that most of the Neo1 residues crucial for establishing PI4P asymmetry are conserved in these mammalian flippases (Extended Data Fig. 4a). Neomycin is poorly absorbed by most human cells and is toxic to only a few tissues, such as kidney cells and inner ear hair cells[7,9]. ATP9A is typically more highly expressed than ATP9B in human tissues, and both are weakly expressed in the kidney (Extended Data Fig. 4c,d). To explore a potential link between ATP9A expression and neomycin sensitivity, we examined ATP9A protein expression levels in HeLa and HEK293 (kidney) cells. Interestingly, ATP9A is highly expressed in HeLa cells but is expressed at low levels in HEK293 cells (Extended Data Fig. 4b,e,f). HeLa cells were resistant to

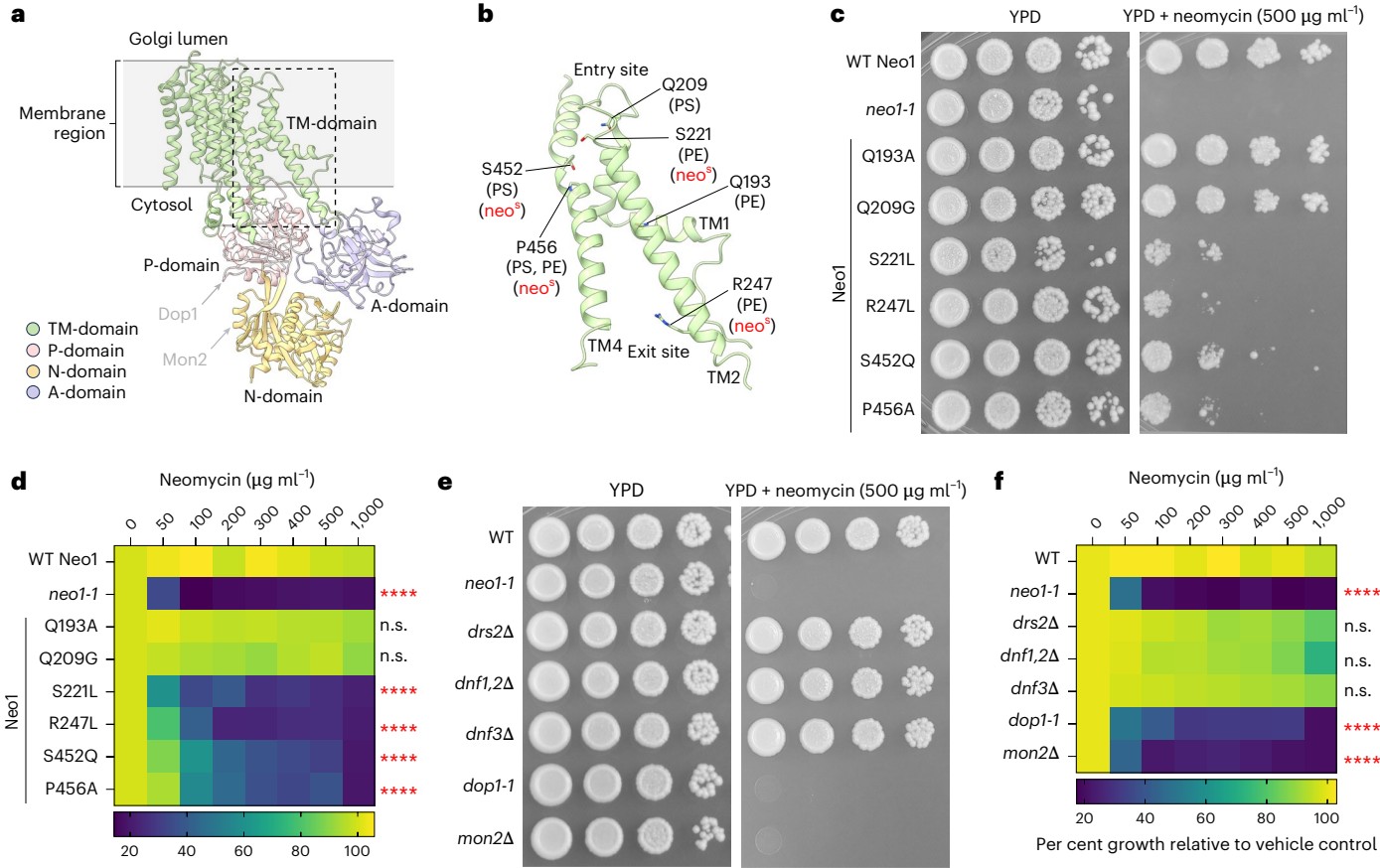

**Fig. 1 | Separation-of-function mutations in Neo1 that cause neomycin sensitivity do not correlate with loss of PS or PE asymmetry. a**, The structure of *S. cerevisiae* Neo1 with the major domains displayed in different colours (PDB ID 9BS1, this study). The dashed box is the substrate translocation pathway enlarged in **b**. **b**, The transmembrane (TM) segments TM1, TM2 and TM4 form the substrate transport pathway of Neo1, and the mutation of residues shown causes exposure of PS and/or PE on the extracellular leaflet of the plasma membrane. **c**, Neomycin sensitivity of Neo1 substrate transport pathway mutants at 26 °C on YPD and YPD neomycin (500 µg ml⁻¹) plates, relative to WT and *neo1-1* (a temperature-sensitive strain displaying a loss of PS and PE membrane asymmetry at this permissive growth temperature). **d**, Neomycin sensitivity in liquid culture of strains expressing the indicated Neo1 variants.

The data represent growth relative to WT cells in the absence of the drug. The mixed model analysis was performed to test the variance and comparisons with WT Neo1 were made with Tukey's multiple comparison analysis (*n* = 3 biologically independent experiments; \*\*\*\**P* < 0.0001). The data are the mean ± s.d. **e,f**, Neomycin sensitivity of *S. cerevisiae* flippase mutants (*neo1-1*, *drs2Δ*, *dnf1,2Δ* and *dnf3Δ*) and strains deficient for Neo1 regulators (*dop1-1* and *mon2Δ*) on YPD + neomycin plates (**e**) and liquid culture (**f**). The data represent the growth relative to WT cells in absence of the drug. A mixed model analysis was performed to test the variance and comparisons with WT Neo1 were made with Tukey's multiple comparison analysis (*n* = 3 biologically independent experiments; \*\*\*\**P* < 0.0001). n.s., not significant. The data are the mean ± s.d.

5,000 µg ml⁻¹ of neomycin, while HEK293 cells displayed moderate sensitivity (Fig. 3a,d). Small interfering RNA (siRNA)-mediated knockdown of ATP9A in HeLa cells rendered them sensitive to neomycin. Similarly, ATP9A depletion in HEK293 cells further increased their neomycin sensitivity (Fig. 3a,d).

Next, we tested whether siRNA-mediated knockdown of ATP9A in HeLa and HEK293 cells would cause exposure of extracellular PI4P. As expected, GFP–SidC_P4C bound to the extracellular leaflet of ATP9A-knockdown cells, and no GFP–PLC_PH binding was detected (Fig. 3b,c,e,f). By contrast, HeLa cells that were fixed and permeabilized stained efficiently with both probes (Extended Data Fig. 2g). Thus, ATP9A knockdown in human cells leads to the exposure of PI4P on the extracellular leaflet. These results indicate that ATP9A plays a crucial role in maintaining PI4P asymmetry and neomycin resistance and suggest that the PI4P flippase activity of Neo1 and ATP9A is conserved from yeast to mammals.

**Cryo-EM captures PI4P at a Neo1 substrate exit site**

ATP hydrolysis by P-type ATPases is typically stimulated by the transported substrate[19,45]. Therefore, we examined whether PI4P can

stimulate ATP hydrolysis by Neo1. Purified Neo1 was incubated with PE (a known substrate)[24,46] or PI4P, revealing a significant stimulation of ATPase activity by both lipids (Fig. 4a). Another yeast P4-ATPase, Drs2, is allosterically activated by PI4P binding interactions near the C-terminus, outside of the substrate translocation pathway. For Drs2, the addition of substrate (PS) or PI4P alone minimally activates ATPase activity, while the combination of both lipids is required for full activation[47–49]. By contrast, the combination of PE and PI4P provided no additional stimulation of Neo1 ATPase activity compared with PE alone (Fig. 4a). This result suggests that PI4P is a Neo1 transport substrate rather than an allosteric activator. We also tested PI4P stimulation of the Neo1 mutants. The S221L and S452Q entry gate mutants surprisingly enhanced the PI4P-stimulated ATPase activity, and the exit gate mutants R247L and P456A reduced this activity (Fig. 4b). This is an unexpected result because these four mutations all cause a similar hypersensitivity to neomycin and loss of PI4P asymmetry (Figs. 1c and 2c).

Major lipid substrates have been observed structurally either at the flippase–substrate entry site (near the extracellular or lumenal leaflet, pre flipping) or at the substrate exit site (near the cytosolic leaflet, post flipping) in the lipid transport pathway[45,50–52]. To investigate how PI4P

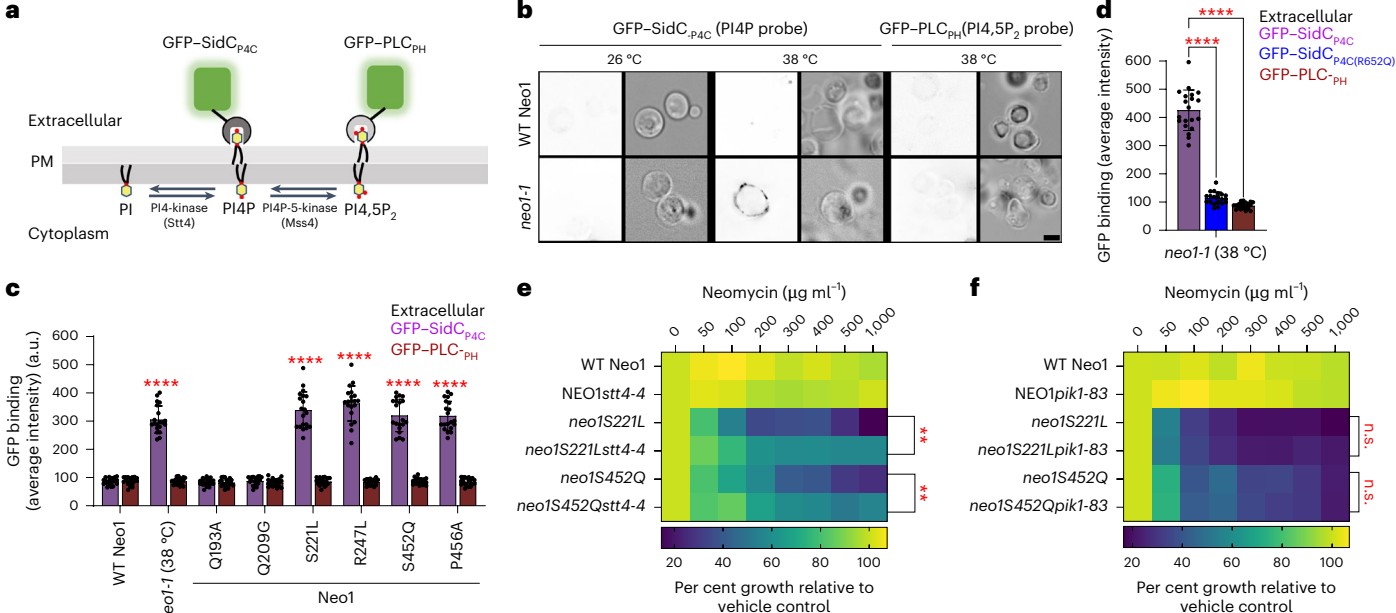

**Fig. 2 | neo1 mutant cells expose PI4P in the plasma membrane extracellular leaflet. a**, PIP synthesis in the cytoplasmic leaflet of the plasma membrane and the use of recombinantly purified GFP–SidC$_{P4C}$ and GFP–PH$_{PLC}$ to probe for external PI4P and PI(4,5)P$_2$, respectively. PM, plasma membrane; PI, phosphatidylinositol. **b**, Inactivation of *neo1-1* at the non-permissive temperature (38 °C) causes PI4P exposure. The fluorescent images were inverted to black signal on white background for clarity. The fluorescence signal intensity of the GFP probe (left) and the differential interference contrast (DIC) panel to display yeast cells (right) are shown. Scale bar, 2 µm. **c**, A quantification of GFP–SidC$_{P4C}$ and GFP–PLC$_{PH}$ binding to the cell surface of cells expressing Neo1 variants. Two-way ANOVA was performed to test the variance and comparisons with WT Neo1 were made with Dunnett's multiple analysis (*n* = 20 cells from

three biologically independent experiments; *P* < 0.0001). The data are the mean ± s.d. **d**, The GFP–SidC$_{P4C}$(R652Q) PI4P binding defective mutant fails to bind *neo1-1* cells. A one-way ANOVA was performed to test the variance and comparisons with WT Neo1 were made with Dunnett's multiple analysis (*n* = 20 cells from three biologically independent experiments; ****P* < 0.0001). The data are the mean ± s.d. **e**, *stt4-4* suppresses the neomycin sensitivity of *neo1S221L* or *neo1S452Q* at 33 °C. **f**, *pik1-83* fails to suppress the neomycin sensitivity of *neo1S221L* or *neo1S452Q* at 34 °C. The data represent growth relative to cells in the absence of the drug. A mixed effect model was performed to test the variance and comparisons with WT Neo1 were made with Tukey's multiple comparison test (*n* = 3 biologically independent experiments; ***P* = 0.0069, 0.0068). n.s., not significant. The data are the mean ± s.d.

binds in Neo1, we reconstituted the purified Neo1 in peptidiscs with added PI4P[53,54] trapped the flippase by BeF$_3^−$ in the substrate-binding E2P state[50] and determined a cryogenic electron microscopy (cryo-EM) structure of the Neo1-PI4P complex at an overall resolution of 3.7 Å (Extended Data Fig. 5 and Supplementary Table 1). We observed a lipid-like EM density that fits well with a modelled PI4P (Fig. 4c). PI4P is at the substrate exit site, representing a post-flipping PI4P in Neo1. Interestingly, the lipid entry site is closed, similar to the closed entry site observed in the Neo1 E2P structure previously determined in detergents (Extended Data Fig. 5). The headgroup of PI4P resides in a large cavity next to TM4, with the terminal phosphate forming hydrogen bonds with a pair of histidine residues (His472 and His476) (Fig. 4d, Neo1-PI4P). Glu475 forms two hydrogen bonds with the PI4P headgroup to stabilize the substrate further. We substituted Glu475, His472 and His476 with alanine, and none of the single mutants showed neomycin sensitivity (Fig. 4e). However, the H472A E475A and H472G E475G double mutants (EH-AA and EH-GG) displayed neomycin sensitivity similar to the Neo1S221L mutants. Thus, the PI4P binding site observed by cryo-EM is important for neomycin resistance.

PI4P binds in the same exit site position and orientation observed for substrate lipid bound in the Dnf1·Lem3 flippase (Extended Data Fig. 6a,b). Thus, the PI4P binding site in Neo1 is within the substrate translocation pathway and is on the opposite side of the membrane domain relative to the PI4P binding site in Drs2 (Fig. 4c). Remarkably, despite the different functions (substrate versus regulatory) and binding sites, the PI4P headgroup is similarly coordinated in Drs2 and Neo1. In Drs2, the binding involves hydrogen bonding with a histidine, a lysine and a tyrosine (Fig. 4c) but with additional H-bonds between the linking phosphate and a duet of tryptophan and arginine residues

from TM10. By contrast, the PI4P binding is weaker in Neo1, perhaps evolved for facile substrate release into the membrane (Fig. 4d). The sequence comparison of PI4P binding site suggests conserved positively charged residues or aromatic residues in other flippases (Extended Data Fig. 7a–c). In sum, these data provide biochemical and structural evidence that PI4P is a translocation substrate of Neo1 rather than an allosteric regulator.

**Transport mechanisms required for PI4P exposure**

Neo1 localizes to the Golgi complex, raising the question of how it controls the asymmetry of PI4P synthesized in the cytosolic leaflet of the plasma membrane. To define the mechanism by which PI4P synthesized in the cytosolic leaflet is exposed on the extracellular leaflet of the plasma membrane, we tested a hypothesis on the basis of the unique metabolism of this PIP proposed to occur at ER–plasma membrane contact sites. After synthesis at the plasma membrane, PI4P can be transported to the ER cytosolic leaflet via the lipid transfer protein Osh6, which in turn acquires PS from the ER and transports it back to the plasma membrane[27–30] (Fig. 5a, WT). PI4P delivered to the ER was presumed to be completely degraded to phosphatidylinositol by the ER-localized Sac1 phosphatase[27,55,56]. However, any lipids present at the ER can undergo rapid flip-flop through an ER scramblase and equilibrate between the lumenal and cytosolic leaflets[17,18,57,58]. Indeed, lumenal PI4P has been detected by freeze-fracture electron microscopy studies of the yeast ER, particularly in *sac1Δ* cells[59]. Therefore, we hypothesized that PI4P could reach the ER lumenal leaflet and travel by vesicular transport to the Golgi, where the Golgi-localized Neo1 would flip PI4P from the lumenal leaflet to the cytosolic leaflet in WT cells. The inability of Neo1 mutants to recognize PI4P would then allow the incorporation of PI4P

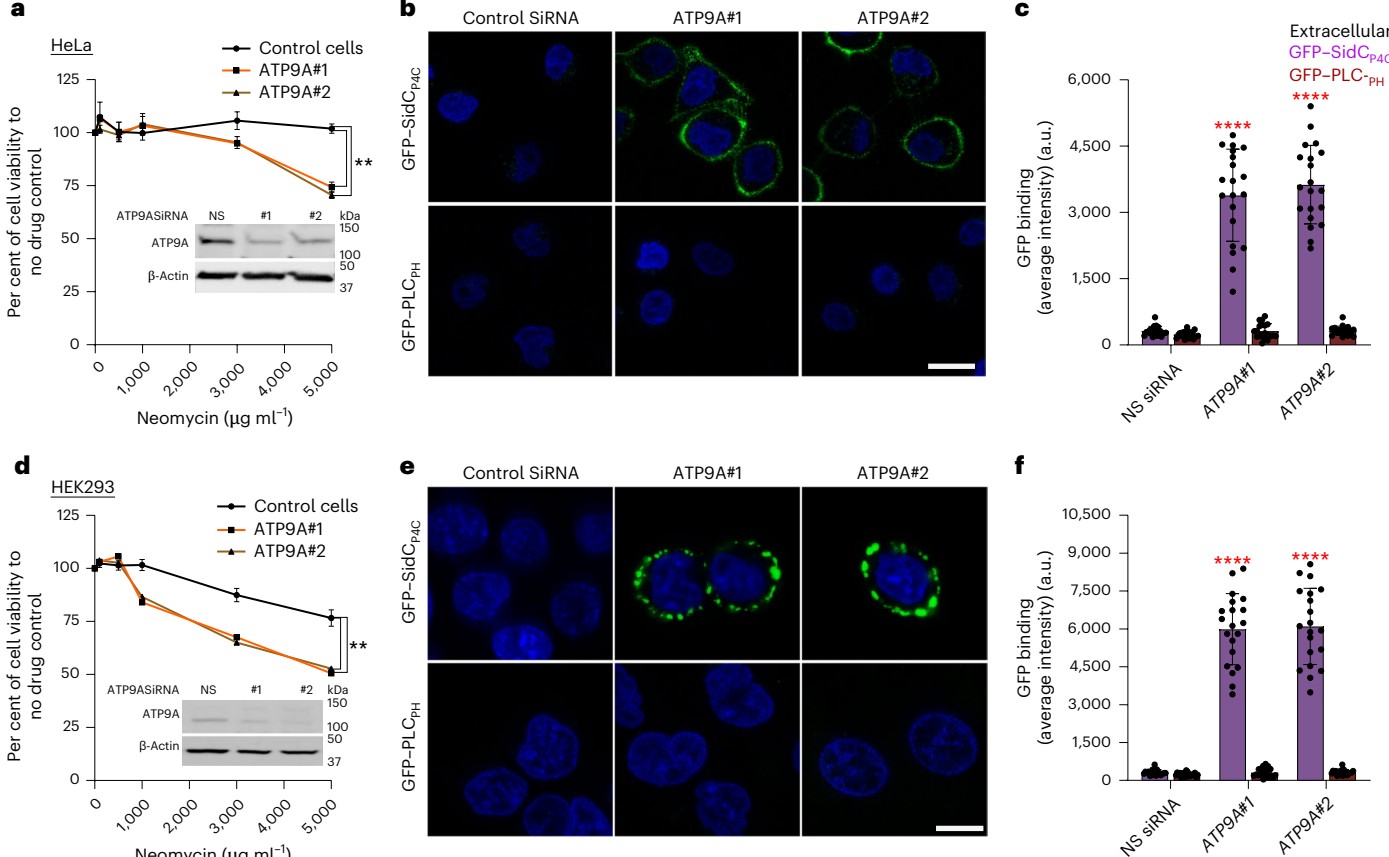

**Fig. 3 | ATP9A-knockdown cells are neomycin sensitive and expose PI4P in the plasma membrane extracellular leaflet. a**, Neomycin sensitivity of WT HeLa cells and ATP9A-knockdown cells. The data represent the per cent of viable cells relative to the viable cells in absence of the neomycin. NS, non-silencing (control) siRNA. A mixed effect model was performed to test the variance and comparisons with control cells were made with Dunnett's multiple comparisons test ($n = 3$ biologically independent experiments; **$P = 0.0054, 0.0025$). The data are the mean ± s.d. **b**, ATP9A-knockdown HeLa cells expose PI4P in extracellular leaflets. Hoechst dye was used as nuclear stain. Scale bar, 10 μm. **c**, A quantification of GFP–SidC$_{P4C}$ and GFP–PLC$_{PH}$ binding to the cell surface of ATP9A-knockdown cells. A two-way ANOVA was performed to test the variance and comparisons with control siRNA treated cells were made with Dunnett's multiple comparisons test ($n = 20$ cells from three biologically independent experiments; ****$P < 0.0001$).

The data are the mean ± s.d. **d**, Neomycin sensitivity of WT HEK293 cells and ATP9A-knockdown cells. The data represent the per cent of viable cells relative to the viable cells in absence of the neomycin. A mixed effect model was performed to test the variance and comparisons with control cells were made with Dunnett's multiple comparisons test ($n = 3$ biologically independent experiments; **$P = 0.0024, 0.0014$). The data are the mean ± s.d. **e**, ATP9A-knockdown HEK293 cells expose PI4P in extracellular leaflets. Scale bar, 10 μm. **f**, A quantification of GFP–SidC$_{P4C}$ and GFP–PLC$_{PH}$ binding to the cell surface of ATP9A-knockdown cells. A two-way ANOVA was performed to test the variance and comparisons with control siRNA treated cells were made with Dunnett's multiple comparisons test ($n = 20$ cells from three biologically independent experiments; ****$P < 0.0001$). The data are the mean ± s.d.

into exocytic vesicles budding from the Golgi, finally resulting in PI4P exposure on the extracellular leaflet of the plasma membrane where it would be accessible to neomycin (Fig. 5a, *neo1* mutant).

To test this hypothesis, we first examined *sac1Δ* cells, which exhibit elevated levels of PI4P at the ER, including the lumenal leaflet[55,59]. Consistently, *sac1Δ* cells displayed neomycin sensitivity similar to *neo1* mutant cells and exposed PI4P in the extracellular leaflet (Fig. 5b–d). However, the extreme sensitivity of *sac1Δ* to neomycin was unexpected because these cells are expressing WT Neo1. This result implies that there is insufficient Neo1 to handle the higher flux of PI4P flowing from the ER to the Golgi lumen in *sac1Δ* cells. To explore this further, we overexpressed WT *NEO1* in *sac1Δ* cells (*sac1Δ/2μ NEO1*), which restored neomycin resistance and reduced PI4P exposure to near WT levels (Fig. 5c,d and Extended Data Fig. 8a). Thus, the combined activities of Sac1 and Neo1 are required to prevent extracellular exposure of PI4P. We attempted to produce *sac1Δ neo1* double mutants to assess their combined deficit on neomycin sensitivity, but the double mutants were either inviable or grew very poorly (Extended Data Fig. 8b). This synthetic lethal genetic interaction supports the importance of Neo1 to PI4P metabolism.

To probe the role of non-vesicular transport by Osh lipid transfer proteins in PI4P exposure and neomycin sensitivity, we created double mutants of *neo1-1* with *osh1Δ–osh7Δ*. Interestingly, only the *osh6Δ* mutation suppressed neomycin sensitivity of *neo1-1*, *neo1-S221L* and *neo1-S452Q* (Fig. 5e and Extended Data Fig. 8c). Osh6 exchanges PI4P from the plasma membrane to the ER cytosol, swapping it with PS, which is then returned to the plasma membrane. Osh4 is the major PI4P/ergosterol transport protein acting at the Golgi where Neo1 and Pik1 are localized[28,56,60], and surprisingly, *osh4Δ* did not influence the neomycin sensitivity of *neo1-1*. These results indicate that PI4P synthesized by Stt4 and transported from the plasma membrane to the ER via the lipid transfer protein Osh6 is crucial for PI4P exposure.

To investigate the involvement of vesicular transport in PI4P exposure and neomycin sensitivity, we generated double mutants of *neo1-1* with temperature-sensitive mutations that perturb COPII vesicle budding from the ER (*sec23* and *sec12*)[61,62], fusion with the Golgi (*sec18-1*)[63,64] or budding of exocytic vesicles from the Golgi (*sec14-1*)[65,66]. At a permissive growth temperature where protein transport is partially perturbed, the *sec* mutations either completely (*sec23-1*

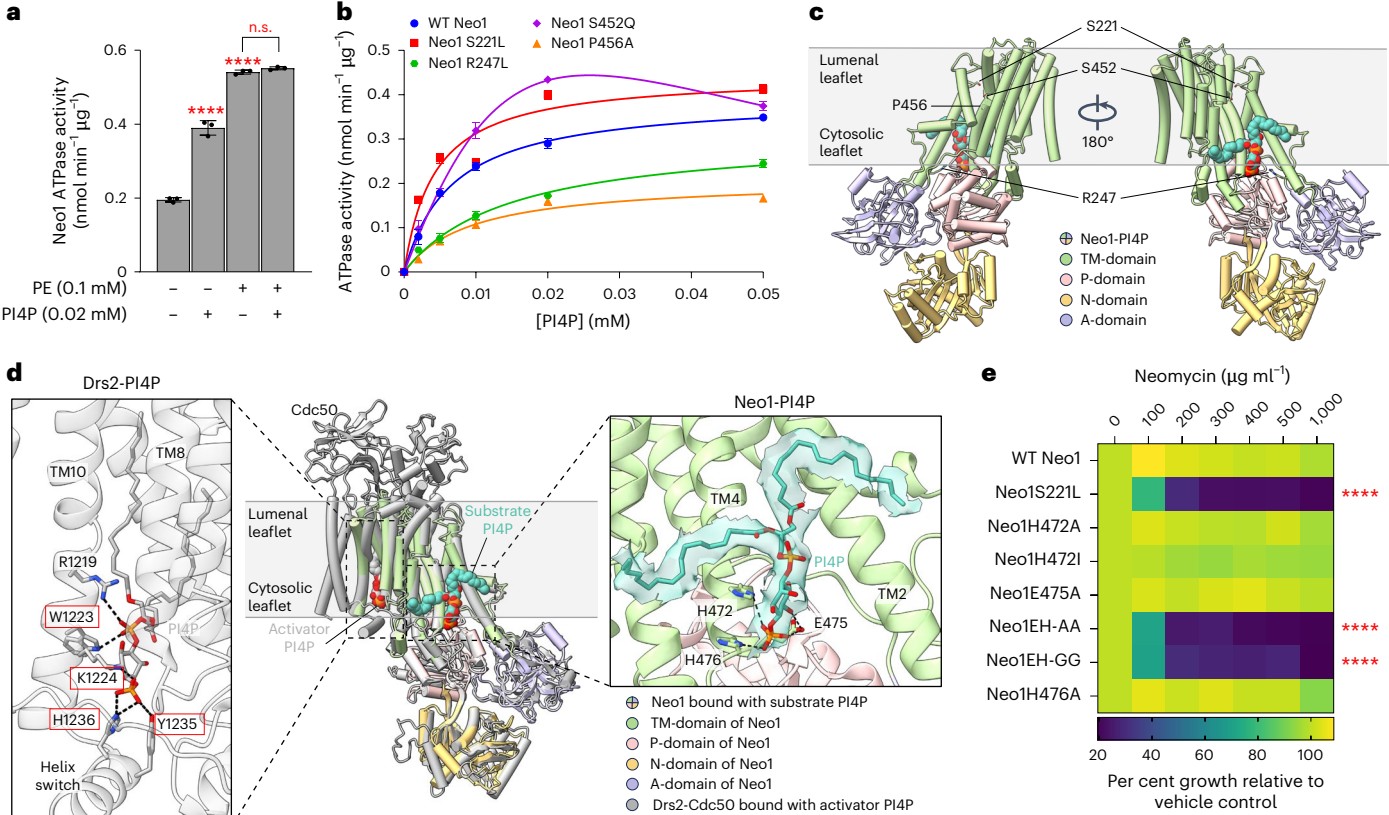

**Fig. 4 | Cryo-EM structure of Neo1-PI4P and comparison of PI4P binding in Neo1 and Drs2. a**, ATP hydrolysis by purified WT Neo1 with or without 0.02 mM PI4P or 0.1 mM PE (substrate) or both PI4P + PE. A one-way ANOVA was performed to test the variance and comparisons with no substrate or PE were made with Dunnett's multiple comparisons test ($n = 3$ biologically independent experiments; ****$P < 0.0001$). n.s., not significant. The data are the mean ± s.d. **b**, ATP hydrolysis activity of WT Neo1 and Neo1 mutants at increasing PI4P concentration. The data points represent the mean ± s.d. from three biologically independent experiments. **c**, Global views of Neo1-PI4P in E2P state with the four key residues in substrate transport pathway shown in sticks and PI4P shown in teal spheres. **d**, Middle: the structure of Neo1-PI4P is superimposed with Drs2-PI4P (grey, PDB ID 6ROJ) and shown in cartoon (root mean square deviation of 4.0 Å). Note the different binding site of PI4P as an activator in Drs2

(grey spheres) and as a substrate in Neo1 (teal spheres). The PI4P binding site residues in Drs2 are highlighted in the red box. Left: an enlarged view of the PI4P binding site in Drs2. Right: an enlarged view of PI4P binding site in Neo1. PI4P and interacting residues are shown in sticks. The hydrogen bonds between the PI4P headgroup and flippase residues are labelled by black dashed lines. The EM density of PI4P in Neo1 is superimposed on the atomic model and shown as a transparent surface. **e**, Mutation in PI4P binding residue causes neomycin sensitivity. HE, H472 E475. The data represent growth relative to WT cells in absence of the drug. A mixed effect model was performed to test the variance and comparisons with WT Neo1 were made with Tukey's multiple comparison test ($n = 3$ biologically independent experiments; ****$P < 0.0001$). The data are the mean ± s.d.

and *sec14-1*) or partially (*sec12-4, sec18-1*) suppressed neomycin sensitivity of *neo1-1*, indicating that vesicular transport from the ER through the Golgi to the plasma membrane is required for PI4P exposure (Fig. 5f).

### PI4P serves as a receptor for neomycin cellular entry

To address how PI4P exposure in the extracellular leaflet confers neomycin sensitivity, we covalently conjugated neomycin with Texas Red (Neo–TR) to visualize how neomycin is taken up (Fig. 6a). Strikingly, there was no significant entry or binding of Neo–TR observed in WT Neo1 cells or the neomycin-resistant Neo1-Q193A and Neo1-Q209G mutants. Conversely, Neo1 mutants that expose PI4P in the extracellular leaflet—namely, S221L, R247L, S452Q and P456A cells—exhibited substantial binding of Neo–TR to the plasma membrane and uptake of the probe to intracellular compartments (Fig. 6b,c). This Neo–TR uptake pattern aligns seamlessly with the exposure of PI4P. Neo–TR localized to the plasma membrane, intracellular punctate structures marked by the late Golgi protein Sec7 and/or endosomal protein Tlg1, and the vacuolar limiting membrane (Fig. 6b,c and Extended Data Fig. 9a,b). This localization pattern implies that extracellular PI4P serves as a binding site for neomycin, and once bound to the plasma membrane, neomycin enters the cells through endocytosis, ultimately reaching the vacuole.

To further investigate the role of endocytosis in Neo–TR entry, we used the endocytosis inhibitor latrunculin A (LatA), known for impeding F-actin formation[67]. The cells were treated with LatA for varying durations, and Neo–TR uptake was measured (Fig. 6d,e). As described earlier, WT Neo1 cells did not exhibit binding to Neo1–TR, showing no fluorescence. In the case of Neo1S221L mutant cells that had not been treated with LatA ($t = 0$), Neo–TR was primarily localized to the plasma membrane with a few intracellular punctate structures. As the duration of LatA treatment progressed to 30 and 60 min, there was an increase in Neo–TR localization at the plasma membrane compared with $t = 0$, and the intracellular puncta and vacuolar staining diminished (Fig. 6d,e). Upon removal of LatA, Neo–TR was internalized and found primarily in intracellular puncta. We also knocked out the endocytic adaptor gene *SLA2* (ref. 68) in the *neo1S221L* background and tested Neo–TR uptake. As expected, we observed increased plasma membrane Neo1–TR levels in the *neo1S221L sla2Δ* strain and reduced internal staining (Extended Data Fig. 9c,d). Surprisingly, the *neo1S221L sla2Δ* strain still displayed sensitivity to neomycin, suggesting that blocking neomycin endocytosis does not protect cells from its toxicity (Extended Data Fig. 9e).

Next, we examined whether Neo–TR could bind to neomycin-sensitive and PI4P-exposing human cells depleted for ATP9A. Control

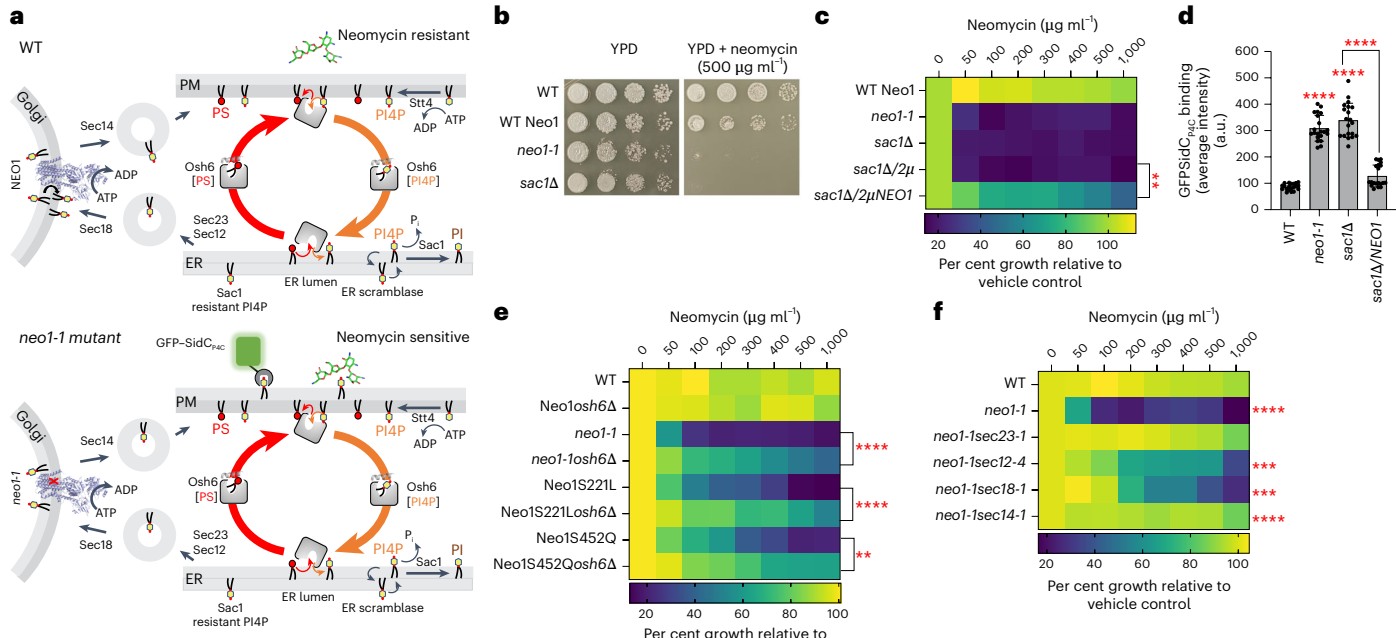

**Fig. 5 | Non-vesicular and vesicular transport is required for PI4P exposure.** **a**, A model for the trafficking of PI4P and the role of Neo1 in maintaining PI4P membrane asymmetry. PM, plasma membrane; P$_i$, inorganic phosphate; PI, phosphatidylinositol; PS, phosphatidylserine. **b**, $sac1\Delta$ cells are neomycin sensitive. **c**, Overexpression of WT $NEO1$ ($2\,\mu\,NEO1$) suppresses $sac1\Delta$ neomycin sensitivity. A mixed effect model was performed to test the variance and comparisons with WT Neo1 were made with Tukey's multiple comparison test ($n = 3$ biologically independent experiments; **$P = 0.0031$). The data are the mean ± s.d. **d**, $sac1\Delta$ cells expose extracellular PI4P and overexpression of $NEO1$ suppresses PI4P exposure. A two-way ANOVA was performed to test the variance, and comparisons with were made with Tukey's multiple comparisons test ($n = 20$ cells from three biologically independent experiments; ****$P < 0.0001$). The data are the mean ± s.d. **e**, Deletion of $OSH6$ ($osh6\Delta$) reduces neomycin sensitivity of $neo1$ mutants. **f**, Reducing the rate of vesicular transport with $sec23$-$1$, $sec12$-$4$, $sec18$-$1$ or $sec14$-$1$ suppresses neomycin sensitivity of $neo1$-$1$ at 26 °C. All the comparisons were calculated via a mixed effect model to test the variance and comparisons with WT Neo1 were made with Tukey's multiple comparison test. The colours represent a comparison with WT, and the pairwise comparisons are shown with brackets and asterisk. The data represent growth relative to WT cells in absence of the drug ($n = 3$ biologically independent experiments; **$P = 0.0029$; ***$P = 0.0005, 0.0009$; ****$P < 0.0001$). The data are the mean ± s.d. For **b**, **c**, **e** and **f**, the cells were grown at 26 °C.

HeLa cells showed no binding of Neo–TR, but ATP9A-knockdown cells showed significant Neo–TR at the plasma membrane and intracellular punctate structures that were primarily excluded from the nucleus (Fig. 6f,g). Interestingly, Neo1–TR significantly stained control HEK293 cells, and ATP9A-knockdown cells showed increased Neo–TR at the plasma membrane, intracellular punctate structures and the nucleus (Fig. 6f,g). In contrast to HeLa ATP9A siRNA cells, Neo1–TR predominantly localized to the nucleus of control and ATP9A-knockdown HEK293 cells. These results support the hypothesis that PI4P provides a cell surface receptor for neomycin cellular uptake.

## Discussion

Here, we describe a mechanism by which the P4-ATPase Neo1 in budding yeast and the orthologous ATP9A in human cells confer neomycin resistance by preventing PI4P secretion to the extracellular leaflet of the plasma membrane. In cells deficient for Neo1/ATP9A, the extracellular PI4P is a receptor that concentrates neomycin on the cell surface and facilitates cellular entry of this aminoglycoside antibiotic. In addition to the previously described PS and PE flippase activity[21,24,46,69], Neo1/ATP9A also functions as a PI4P flippase, transporting PI4P from the lumenal leaflet of the Golgi to the cytosolic leaflet. These discoveries provide a mechanistic explanation for the cell-type-specific toxicity of neomycin and have broader implications for PIP homoeostasis in antibiotic resistance, cellular physiology and human pathology[13,70].

In humans, $ATP9A$ homozygous recessive mutations cause neurodevelopmental defects leading to postnatal microcephaly, intellectual disability, attention deficit hyperactivity disorder and hypotonia[71–73]. Neo1 and ATP9A play essential roles in protein trafficking between the Golgi and endosomes, and perturbation of these pathways may cause these neurodevelopmental defects in patients[44,71,74]. It is also possible that the loss of plasma membrane asymmetry and perturbation of PIP metabolism contribute to neuronal dysfunction[75]. The findings presented here have significant implications for understanding the toxicity of aminoglycosides such as neomycin. The cell-type-specific expression of ATP9A and ATP9B may explain why certain tissues, such as kidney cells and inner ear hair cells, are particularly susceptible to neomycin-induced damage (Extended Data Fig. 4c–f). The expression level and activity of human Sac1 (SACM1L) may also contribute to neomycin sensitivity. In addition, identifying PI4P as a neomycin receptor opens avenues for developing strategies to mitigate aminoglycoside toxicity, such as targeting PI4P exposure or modulating flippase activity.

Neomycin binds to extracellular PI4P[9,10] and is subsequently endocytosed into the cell, as demonstrated by the uptake of Neo–TR in Neo1 and ATP9A deficient cells. This uptake is mediated by endocytosis in yeast, as inhibition of the endocytic pathway reduces Neo–TR internalization. These observations suggest that PI4P serves as a cell surface receptor for neomycin, facilitating its entry into cells and subsequent accumulation in endocytic and vacuolar compartments. Surprisingly, inhibition of endocytosis reduces the Neo–TR internalization but does not affect yeast cytotoxicity, suggesting that neomycin can directly cross the plasma membrane to intoxicate the cell. The difference in the localization pattern of Neo–TR between HeLa and HEK293 cells also suggests different cytotoxic mechanisms for these two cell types. Although the precise mechanism for how the polycationic neomycin crosses a lipid bilayer and inhibits growth remains to be determined, binding to exposed PI4P is a critical determinant of cytotoxicity.

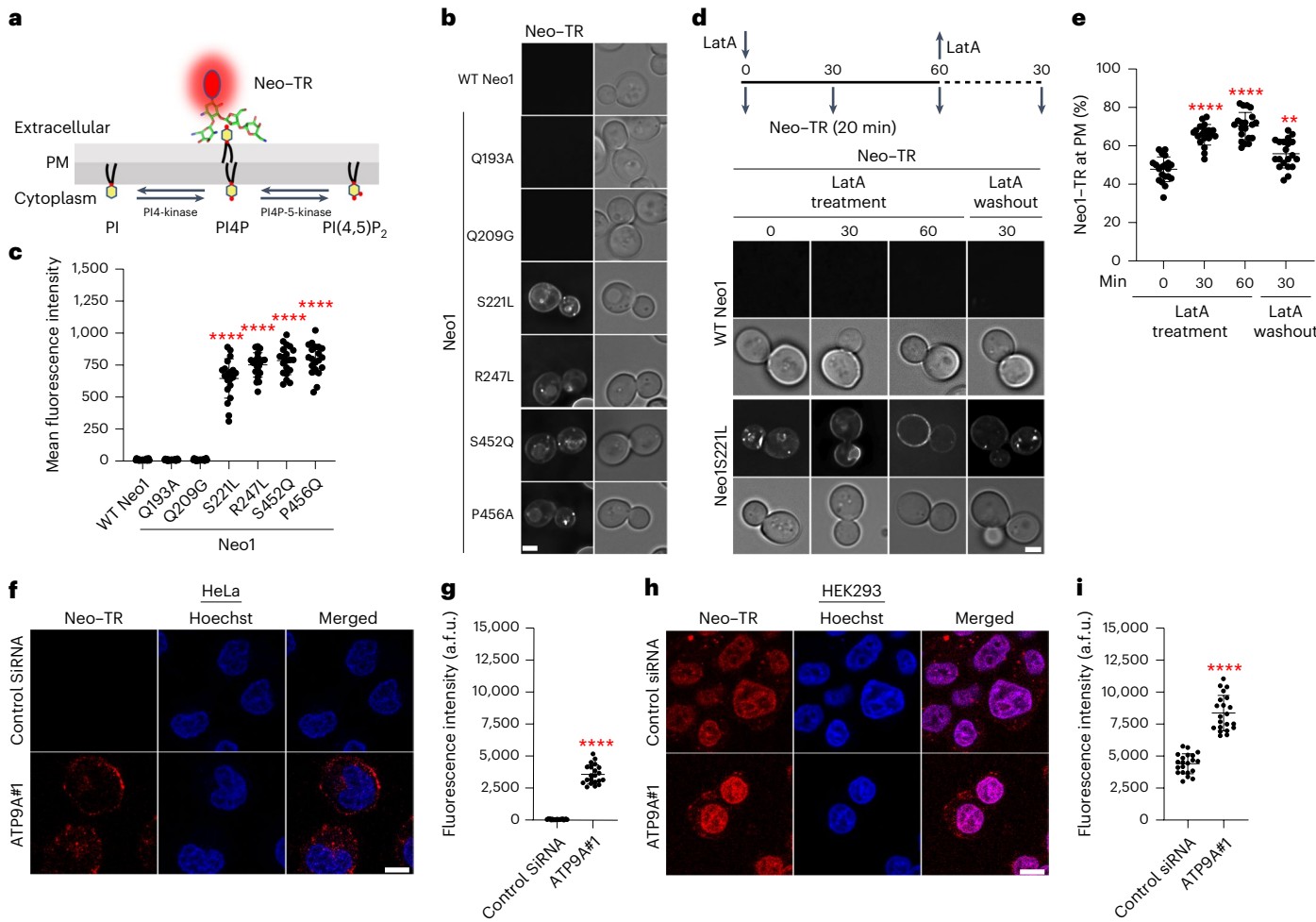

**Fig. 6 | Neomycin binds to exposed PI4P and is endocytosed. a**, Use of Neo–TR to probe PI4P exposed on the extracellular leaflet of the membrane. **b**, Neo–TR binds neomycin-sensitive *neo1* mutants. Scale bar, 2 µm. **c**, A quantification of mean fluorescence intensity of cells stained with Neo–TR. A one-way ANOVA was performed to test the variance and comparisons with WT Neo1 were made with Dunnett's multiple comparisons test (*n* = 20 cells from three biologically independent experiments; ****$P < 0.0001$). The data are the mean ± s.d. **d**, LatA blocks Neo–TR internalization. Scale bar, 2 µm. **e**, A quantification of Neo–TR fluorescence intensity at the plasma membrane (PM). For all quantification, the data from ~20 cells from three independent experiments were obtained and analysed. A one-way ANOVA was performed to test the variance and comparisons with different timepoints were made with Tukey's multiple comparisons test (*n* = 20 cells from three biologically independent experiments; **$P = 0.0014$; ****$P < 0.0001$). The data are the mean ± s.d. **f**, Neo–TR binds to the ATP9A-knockdown HeLa cells. Hoechst dye was used as a nuclear stain. **g**, A quantification of total Neo–TR fluorescence intensity of cells (*n* = 20 cells from three biologically independent experiments). Scale bar, 10 µm. **h**, Neomycin-sensitive HEK293 and ATP9A-knockdown cells bind Neo–TR. Scale bar, 10 µm. **i**, A quantification of total Neo–TR fluorescence intensity (*n* = 20 cells from three biologically independent experiments). For **g** and **i**, a two-tailed unpaired *t*-test was performed to test the variance and comparisons with control cells (*n* = 20, mean ± s.d. ****$P < 0.0001$).

We uncovered a pathway for exposure of PI4P in the extracellular leaflet of the plasma membrane of *neo1* mutants that entails both non-vesicular and vesicular transport of this lipid. This pathway starts with the synthesis of PI4P in the cytosolic leaflet of the plasma membrane by Stt4 (ref. 37) and its Osh6-dependent, non-vesicular transport to the ER. Of the seven Osh proteins in budding yeast, only Osh6 appears to contribute significantly to the exposure of PI4P. Osh7 is thought to act redundantly with Osh6 for PS transport from the ER to the plasma membrane, but we did not detect a role for Osh7 in this pathway. However, it is possible that Osh7 overexpression in the *osh6Δ* background would reveal a contribution to the PI4P flux. Similarly, PI4P produced at the Golgi by Pik1 and potentially transported to the ER by Osh4 did not contribute to its extracellular exposure. The biosynthetic ER scramblase probably mediates the penetration of PI4P into the ER lumen, where it is protected from degradation by Sac1. Finally, COPII-dependent ER to Golgi transport and Sec14-dependent Golgi to plasma membrane vesicular transport deliver PI4P to the extracellular leaflet of the plasma membrane. The conservation of this pathway in human cells, where knockdown of the Neo1 orthologue ATP9A leads to extracellular PI4P exposure and neomycin sensitivity, underscores the evolutionary importance of this mechanism.

PI4P exposure in *neo1* mutants and the ATP9A-knockdown cells suggest that a substantial flux of PI4P escapes degradation by Sac1 and enters the ER lumen[59]. This raises the possibility that PI4P could regulate proteins of the ER, Golgi and plasma membrane by interacting with lumenal/extracellular domains. In mammalian cells, extracellular PI4P could also mediate cell–cell signalling events[13,70,76]. In addition, the loss of Sac1 greatly increases the undercurrent of PI4P in the secretory pathway, overwhelming the capacity of Neo1 to pump this lipid to the cytosolic leaflet. Sac1 is regulated by nutrient availability, and glucose deprivation causes Sac1 redistribution to the Golgi complex[77,78]. Reduction of Sac1 at the ER could increase the flux of PI4P entering the ER lumen and travelling through the secretory pathway. Determining if lumenal/extracellular PIPs contribute to nutrient acquisition or energy metabolism will be interesting.

The cryo-EM structure of Neo1 in the E2P state with bound PI4P provides further insight into substrate selection and transport mechanisms by P4-ATPases. From a collection of mutants targeting the substrate translocation pathway, we initially identified four mutations that caused neomycin sensitivity and PI4P exposure. Two of these mutations are in the entry gate substrate-binding site (S221L and S452Q) as defined in related P4-ATPases, one is near the 'hydrophobic gate' separating the entry and exit sites (P456A), and one is near the exit gate substrate-binding site (R247L). The cryo-EM structure shows PI4P is in the exit gate substrate-binding site and identified residues critical for headgroup coordination (H472, H476 and E475). Consistent with a flippase–substrate interaction, the *neo1-H472A E475A* double mutant causes a loss of PI4P asymmetry and hypersensitivity to neomycin. As previously observed, the entry site is closed in the E2P structure and is presumably opened in vivo through interactions with Dop1 and Mon2. P4-ATPases do not require lipid substrate for autophosphorylation (E1 to E1P to E2P); instead, substrates induce dephosphorylation (E2P to E2) to stimulate overall ATPase activity. PI4P-stimulated purified Neo1 ATPase activity as expected for a transport substrate. Interestingly, the P456A and exit gate R247L mutations diminish, but the entry gate S221L and S452Q mutations enhance the ability of PI4P to stimulate ATPase activity. This unexpected behaviour of the entry gate mutations suggests they are opening the entry gate binding site to enhance substrate-stimulated ATPase activity. S221L and S452Q may disrupt PI4P transport by allowing PS or PE to outcompete PI4P at the entry gate binding site in vivo. Further studies are needed to fully understand the biochemical consequences of these mutations.

In summary, this study sheds light on the broader role of extracellular PIPs in cellular signalling and membrane dynamics. The exposure of PI4P on the cell surface, as observed in Neo1 and ATP9A deficient cells, may have implications for other physiological processes, including cell signalling, membrane trafficking and pathogen interactions[13,70,71]. Further investigation into the regulation and function of extracellular PIPs will undoubtedly yield insights into how these signalling lipids regulate cell physiology and impact disease mechanisms.

## Online content

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

## Methods

### Yeast strains and plasmid construction

The strains and plasmids used in the study are listed in Supplementary Table 2. All yeast culture reagents were purchased from Sigma-Aldrich and BD Scientific, and the strains were grown in yeast extract, peptone, dextrose (YPD) or minimal selective media. Yeast transformation was performed using the LiAC-PEG method[79]. Single and double deletion strains were generated using the homologous recombination method[80]. Open reading frames were replaced with the kanMX6, CloNAT or His3MX6 cassettes and confirmed by PCR. Neomycin was purchased from Gibco. The colony growth assays were performed by spotting 1 optical density at 600 nm ($OD_{600}$) unit and tenfold serial dilutions on YPD and YPD + neomycin (500 μg ml$^{-1}$) plates. The plates were incubated for 2–3 days at 26 °C or indicated temperature before capturing the images. The images of yeast colonies are representative of three biological replicates, each derived from independently isolated strains with the same genotype. DNA constructs and mutations were generated through Gibson assembly (New England Biolabs) and quick-change mutagenesis (Agilent).

### Cell culture and siRNA knockdown

The HeLa cell lines were obtained as gifts from Hye-Won Shin Lab (Kyoto University), and the HEK293 cells were obtained as a gift from Lars Plate (Vanderbilt University). The cells were cultured at 37 °C with 5% $CO_2$ in DMEM (Gibco) with 10% FBS. The cell lines were routinely monitored for mycoplasma contamination. The predesigned siRNA duplexes against human ATP9A were purchased from Thermo Fisher Scientific with following sequences: ATP9A SiRNA#1: GAUUCCUCAUCAUUGGGUAtt (s19607) and ATP9A SiRNA#2: CGUGGGAACUUUUACCCGAtt (s19609). Control SiRNA Silencer Select Negative Control No. 1 siRNA was also purchased from Thermo Fisher Scientific (4390844). The HeLa cells were seeded on six-well plates at a density of $1.5 \times 10^5$ cells per well. The cells were transfected with 30 nM siRNA using Lipofectamine RNAiMAX Reagent (Invitrogen-13778150). The knockdown efficiency was validated by western blots after the siRNAs were transfected for 48 h using Anti-ATP9A antibody.

### Protein extraction and immunoblotting

The cells were washed with phosphate-buffered saline (PBS) and scraped from the plate in RIPA buffer (Sigma-Aldrich R0287) plus protease and phosphatase inhibitors. The cell lysates were transferred into a clean microcentrifuge tube and vortexed for 20 s at high speed. The lysates were centrifuged at 14,000$g$ for 15 min, and the supernatant was collected in a microcentrifuge tube. The protein concentration was measured using Pierce BCA protein assay Kit (Thermo Scientific-23225). A total of 20 μg of protein samples were loaded on a 4–20% gradient SDS–PAGE gel (Bio-Rad, cat. no. 4561095) and transferred to PVDF membranes. ATP9A was probed using primary antibody rabbit anti-ATP9A antibody (ab234873, 1:1,000) and secondary antibody anti-rabbit HRP (Promega W4011, 1:10,000). The loading control β-actin was probed using primary mouse anti-β-actin antibody (CST#3700, 1:10,000) and secondary anti-mouse HRP (Promega W4021,1:10,000) antibody.

### Neomycin sensitivity assay for yeast cells

The cells grown to the mid-log phase were diluted to 0.1 $OD_{600}$ in fresh YPD medium, and 100 μl of the resulting cells were dispensed into each well of a 96-well plate, with or without neomycin, in an additional 100 μl of YPD. Neomycin dilutions were calculated to achieve final concentrations in a total well volume of 200 μl. The plates were then incubated at 26 °C for 20 h. The cell concentrations were measured in $OD_{600}$ ml$^{-1}$ using a Multimode Plate Reader Synergy HT (Biotek). Growth, relative to the vehicle control (no drug), was considered as 100% growth. All values represent averages from at least three independent biological replicates, mean ± standard deviation (s.d.). Statistical analysis were determined using a mixed-effects analysis model followed by a Dunnett's or Tukey's multiple comparison analysis.

### Neomycin sensitivity assay for mammalian cells

The HeLa cells or HEK293 cells were treated with respective siRNA for 48 h in six-well plates. The cells were collected, and approximately 5,000 cells were seeded on 96-well plates in 100 μl of complete media (DMEM + FBS + P/S) and allowed to attach dish for 24 h. After 24 h, 100 μl of complete media was replaced with the 100 μl of neomycin containing media (DMEM + FBS + neomycin) and incubated for 48 h at 37 °C. The cell viability was measured using WST-1 Assay Kit (ab65473). Briefly 10 μl of WST-1 reagent was added to each well and incubated for 4 h at 37 °C for formazan formation by viable cells. The amount of formazan produced by viable cells was quantified using Multimode Plate Reader Synergy HT (Biotek) at 440 nm. The viable cells at the no drug condition were considered as 100% growth. All values represent averages from at least three independent biological replicates, mean ± s.d. The statistical analysis were determined using a mixed-effects analysis model followed by Dunnett's multiple comparison analysis.

### GFP–SidC$_{P4C}$ protein expression and purification

Recombinant GFP–SidC$_{P4C}$ and GFP–SidC$_{P4C(R562Q)}$ was expressed and purified from $Escherichia$ $coli$[38]. Briefly, $E.$ $coli$ Rosetta2 pLysS strains containing the expression plasmids were cultured in Luria–Bertani media supplemented with 50 μg ml$^{-1}$ kanamycin and 25 μg ml$^{-1}$ chloramphenicol until reaching mid-log phase. The protein expression was induced overnight at 18 °C by adding 0.1 mM isopropyl-β-ᴅ-thiogalactopyranoside. The collected cells were resuspended in a buffer composed of 20 mM Tris–HCl (pH 7.5) and 150 mM NaCl and lysed using a cell disrupter (Constant Systems). The soluble fractions were obtained through centrifugation at 38,000$g$ for 30 min at 4 °C and were then incubated with TALON metal affinity resins (Takara-635502) for 1 h at 4 °C. The protein-bound resins were thoroughly washed with the lysis buffer. The protein was eluted by stepwise gradient (15, 30, 40, 50, 60, 180, 180 and 240 mM imidazole), and the fractions containing the protein were analysed by SDS–polyacrylamide gel electrophoresis and peak fractions were pooled together. An Amicon Ultra-15 centrifugal filter unit (Millipore, UFC903008) was used to dialyse into an imidazole-free buffer containing 20 mM Tris, pH 7.5 and 50 mM NaCl.

### GFP–PLC$_{PH}$ expression and purification

Recombinant GFP–PLC$_{PH}$ was expressed and purified from $E.$ $coli$ BL21 (DE3)[39]. The $E.$ $coli$ strain expressing GFP–PLC$_{PH}$ was grown in Terrific Broth containing 4 ml l$^{-1}$ glycerol, 50 μg ml$^{-1}$ kanamycin and 0.8% (wt/vol) ᴅ-glucose. The cells were induced with 0.1 mM isopropyl-β-ᴅ-thiogalactopyranoside at 0.6 OD for 20 h at 23 °C. The cells were collected and resuspended in lysis buffer containing 25 ml of 300 mM NaCl, 40 mM Tris, 15 mM imidazole, 5% (vol/vol) glycerol, pH 7.4 (10 °C), 5 mM β-mercaptoethanol, protease inhibitors and DNase. The cells were lysed using a cell disrupter (Constant Systems), and the lysate was centrifuged at 27,000$g$ for 20 min. The supernatant was incubated overnight with TALON metal affinity resin (Takara-635502) at 4 °C. The resin was washed twice with 10–15 mM imidazole in lysis buffer and eluted using a stepwise imidazole gradient. The peak fractions analysed by SDS–polyacrylamide gel electrophoresis were pooled and dialysed (Thermo Fisher, 66380; Slide-A-Lyzer Dialysis Cassettes) into imidazole-free buffer (150 mM NaCl, 20 mM Tris, 5% (vol/vol) glycerol and 0.5 mM β-mercaptoethanol, pH 7.4).

### GFP–SidC$_{P4C}$ and GFP–PH$_{PLC}$ binding assay for yeast cells

The yeast strains were grown in YPD media, and the cells were collected while in log phase (0.5 $OD_{600}$ ml$^{-1}$). The cells were washed once in 50 mM Tris–Cl, pH 7.4, and resuspended at 50 $OD_{600}$ ml$^{-1}$ in 50 mM Tris–Cl, pH 7.4, containing 1.4 M sorbitol. The cells were then converted to spheroplasts during a 45-min incubation with 1 μl of Zymolyase-100T (10 mg ml$^{-1}$ in water; Seikagaku) per 10 $OD_{600}$ units of

cells. The spheroplasts were resuspended at 1 $OD_{600}$ cells ml$^{-1}$ in buffer (50 mM Tris, pH 7.4, 50 mM NaCl with 1.4 M sorbitol). A total of 2 µl of GFP–SidC$_{P4C}$ (12 µg ml$^{-1}$) was added to 200 µl spheroplast suspension, and the mixture was incubated for 20 min on ice. The cells were washed two times with buffer (50 mM Tris, pH 7.4, 50 mM NaCl with 1.4 M sorbitol), and the cells were then mounted on glass slides and observed immediately at room temperature. The images were acquired using a DeltaVision Elite Imaging System equipped with a 100×, 1.4 NA oil immersion objective lens followed by deconvolution using softWoRx software (v7.0.0; GE Healthcare). The images were analysed and fluorescence intensity was quantified using ImageJ (1.54g). For GFP–PLC$_{PH}$ binding 1 µl (20 µg ml$^{-1}$) of purified protein was incubated with 200 µl spheroplast suspension, and the mixture was incubated for 20 min on ice. The cells were washed two times with buffer (50 mM Tris, pH 7.4, 50 mM NaCl with 1.4 M sorbitol) and mounted on the glass slide. The images were acquired at room temperature and GFP–PLC$_{PH}$ binding was quantified using ImageJ.

### GFP–SidC$_{P4C}$ and GFP–PH$_{PLC}$ binding assay for mammalian cells
The HeLa cells were treated with respective siRNA for 48 h in six-well plates. The cells were collected and approximately 50,000 cells were seeded on 35-mm glass-bottom MatTek Dishes (P35G-1.5-14-C) and allowed to attach to the dish for 24 h. After 24 h, the cells were washed with 1 ml of 1× PBS and Hoechst stain (33342, 1:1,000) in PBS was added to the cells and incubated for 15 min at 37 °C. The cells were washed two times with PBS and 10 µl of GFP–SidC$_{P4C}$ (12 µg ml$^{-1}$) or 5 µl (20 µg ml$^{-1}$) of GFP–PLC$_{PH}$ in 1 ml PBS was added to 1 ml ice-cold PBS, and the mixture was incubated for 40 min on ice. The cells were washed two times with 1× PBS. The images were acquired using Inverted LSM880-Airyscan (Zeiss) microscope equipped with a 63×, 1.4 NA oil immersion objective lens followed by deconvolution using Zen Desk software. The images were analysed and GFP–SidC$_{P4C}$ or GFP–PLC$_{PH}$ binding was quantified using ImageJ (1.54g). The HeLa cells were permeabilized and stained as previously described[40]. HeLa cells were cultured in a 6-well uncoated glass dish (10 mm diameter) to 80% confluence in complete media. After removing the media, the cells were washed with 1 ml of 1× PBS. Hoechst stain (33342, 1:1,000) in PBS was then added and incubated for 15 min at 37 °C. Following two PBS washes, the cells were fixed with 4% paraformaldehyde and 0.2% glutaraldehyde at room temperature for 20 min. The cells were then washed twice with 50 mM ammonium chloride and incubated for 20 min. The six-well plates were placed on ice and washed twice with ice-cold PIPES buffer. Fresh staining solution (5% BSA, 0.5% (vol/wt) saponin, GFP–SidC$_{P4C}$ or GFP–PLC$_{PH}$) was prepared, and 200 µl of this solution was added to each sample and incubated on ice for 45 min. After staining, the cells were washed three times with ice-cold PIPES buffer. Post fixation was performed with 2% paraformaldehyde in PBS on ice for 10 min, followed by 10 min at room temperature. The cells were then washed with 50 mM NH$_4$Cl in PBS, rinsed with water, mounted and sealed using ProLong Diamond Antifade.

### Neomycin–TR labelling
Neomycin was conjugated to Texas Red[81]. Neomycin sulfate hydrate (Sigma-Aldrich) was used at a final concentration of 115.6 mg ml$^{-1}$. A total of 115.6 mg neomycin sulfate hydrate was initially resuspended in 500 µl of deionized water, to which 176 µl of 0.5 M K$_2$CO$_3$ at pH 9.0 and 120 µl of Texas Red-X-succinimidyl ester (Life Technologies) dissolved in dimethylformamide at 2.5 mM was added. The reaction volume was adjusted to 1 ml with deionized water and the solution was incubated overnight at 4 °C to complete the conjugation reaction.

### Neomycin–TR uptake assay for yeast cells
The cells were grown up to mid-log phase in rich media YPD. The cells were collected and washed three times in SD media. 2 µl of Neo–TR from stock solution was added to 50 µl of cells and incubated on ice for 20 min. The cells were washed two times with fresh media and resuspended in fresh media. The images were collected using Delta Vision Elite Imaging System equipped with a 100×, 1.4 NA oil immersion objective lens followed by deconvolution using softWoRx software (GE Healthcare Cytiva). The fluorescence intensity was measured using ImageJ. For the LatA experiments, the cells were treated with Neo–TR for 20 min and treated with LatA for 30 and 60 min. The cells were washed and resuspended in fresh media. The cells were kept on ice until mounted on the coverslip.

### Neomycin–TR uptake assay for mammalian cells
The HeLa cells and HEK293 cells were treated with respective siRNA for 48 h in six-well plates. The cells were collected, and approximately 50,000 cells were seeded on 35-mm glass-bottom MatTek Dishes (P35G-1.5-14-C) and allowed to attach dish for 24 h. After 24 h, the cells were washed with 1 ml of 1× PBS and Hoechst stain (33342, 1:1,000) in PBS was added to the cells and incubated for 15 min at 37 °C. The cells were washed two times with PBS and 20 µl of Neo–TR from stock solution was added to 1 ml ice-cold PBS, and the mixture was incubated for 30 min on ice. The cells were washed two times with 1× PBS. The images were acquired using Inverted LSM880-Airyscan (Zeiss) microscope equipped with a 63×, 1.4 NA oil immersion objective lens followed by deconvolution using Zen Desk software. The images were analysed and Neo–TR fluorescence intensity was quantified using ImageJ (1.54g).

### Molecular cloning
The gene sequence encoding Neo1 from *S. cerevisiae* was PCR-amplified from genomic DNA of *S. cerevisiae* BY4741 obtained from Horizon Discovery with a pair of primers 5′- GATTACAAGGATGAC-GACGATAAGCCTAACCCTCCTTCATTTAAGTCACATAAGC-3′/5′-ATTAAGAGCTCATGGAGTGGCAAACTCTTGCAC-3′. The gene was inserted using ligation-independent cloning into a vector modified from pJK369 (plasmid #73594) acquired from Addgene for the expression of Neo1. After DNA sequence verification (Plasmidsaurus), the *S. cerevisiae* strain BY4741 *Δpep4::kanMX6* (Horizon Discovery) was transformed with the constructed vector using Quick and Easy Yeast Transformation Mix (Takara Bio). The expressed Neo1 had an N-terminal FLAG-tag.

### FLAG-Neo1 expression and purification
The cells were grown in 1.2 l of SD-Ura (Synthetic drop-out without uracil) medium (20 g l$^{-1}$ glucose, 6.7 g l$^{-1}$ yeast nitrogen base without amino acids and 1.92 g l$^{-1}$ yeast synthetic drop-out medium supplements without uracil) at 30 °C, shaking at 250 rpm, until exhaustion of glucose in the culture monitored using a Contour Next One blood glucose metre (Ascensia Diabetes Care). The gene expression was induced with a final concentration of 2% galactose in 4.8 l of YP medium (10 g l$^{-1}$ yeast extract and 20 g l$^{-1}$ peptone), and the cells were grown for an additional 16 h at 30 °C, 220 rpm. The cells were collected by centrifugation at 10,000$g$ at 4 °C for 5 min, and the cell pellet was washed and resuspended in protein buffer (20 mM HEPES, pH 7.4, 150 mM NaCl and 1 mM MgCl$_2$), then dripped into liquid nitrogen and finally stored at −80 °C before cell lysis. The cells were lysed using a SPEX Freezer/Mill 6875 (SPEX SamplePrep) with the following settings: precool: 1 min, run time: 2 min, cool time: 1 min, cycles: 5 and rate: 15 cycles per second. The milling was repeated one time, then the lysed cells were resuspended in protein buffer also containing 1× ProBlock Gold Yeast/Fungi Protease Inhibitor Cocktail (GoldBio) and 10 µl Benzonase Nuclease HC (EMD Millipore). The resuspension was further incubated at 4 °C for 30 min with gentle stirring. After centrifugation at 6,000$g$ for 15 min at 4 °C, the supernatant was filtered through an Econo-Column chromatography column with a 20-µm pore size bed support (Bio-Rad). The resulting protein solution was ultracentrifuged at 137,900$g$ for 2 h at 4 °C, and the membrane pellet was resuspended in resuspension buffer (20 mM Tris, pH 7.4, 500 mM NaCl,

10% (w/v) glycerol and 1 mM MgCl$_2$) and homogenized with a Dounce homogenizer. The resulting membrane homogenate was treated with final concentrations of 1% $n$-dodecyl-β-D-maltopyranoside, 0.1% cholesteryl hemisuccinate Tris salt (CHS) and 1 mM 4-(2-aminoethyl) benzenesulfonyl fluoride hydrochloride and incubated at 4 °C for 1 h with constant stirring. The extracted membrane proteins were collected after ultracentrifuge at 137,900$g$ for 1 h at 4 °C and then mixed with 2 ml bed volume of pre-equilibrated anti-DYKDDDDK G1 affinity resin (GenScript) and incubated at 4 °C for 3 h with stirring. The mixture was then transferred to an Econo-Column chromatography column (Bio-Rad) at 4 °C. After flow-through collection, the column was washed with protein buffer with detergents (0.01% lauryl maltose neopentyl glycol (LMNG) and 0.001% CHS) containing 0 mM, 0.25 mM and 0 mM ATP in sequence in ten bed volumes. Two strategies have been reported in the literature for removing endogenously bound lipids in a lipid flippase, either via excessive column wash[51] or by wash with ATP during purification[82]. We used the later approach by adding an ATP wash step. Finally, the column was developed with three bed volumes of protein buffer with detergents (0.01% LMNG and 0.001% CHS) and 0.5 mg ml$^{-1}$ DYKDDDDK peptide. The eluate was collected and concentrated to 0.5 ml with a 50 kDa cutoff centrifugal filter (EMD Millipore) at 4,000$g$ at 4 °C. Finally, the protein concentrate was further purified with a Superose 6 Increase 10/300 GL column (Cytiva) at a flow rate of 0.5 ml min$^{-1}$ in final protein buffer with detergents (0.0025% LMNG and 0.00025% CHS). The peak fractions of the target protein were pooled and concentrated before prompt use or flash-freezing in liquid nitrogen and storage at −80 °C.

## Peptidisc reconstitution of Neo1
Concentrated Neo1 at 5.325 mg ml$^{-1}$ was first incubated with 0.5 mM PI4P dissolved in 4 mM sodium cholate at 4 °C for 3 h, and then, 300 μl pH-adjusted peptidisc solution containing 0.5 mg peptidisc peptide (Peptidisc Biotech) was added and incubated further on ice for 30 min. Neo1 in peptidiscs was further purified with a Superose 6 Increase 10/300 GL column (Cytiva) at a flow rate of 0.5 ml min$^{-1}$ in final protein buffer without detergents (20 mM HEPES, pH 7.4, 150 mM NaCl and 1 mM MgCl$_2$). The peak fractions of the target protein were pooled and concentrated to 1.511 mg ml$^{-1}$.

## Cryo-EM grid preparation and data collection
For the E2P state of peptidisc-reconstituted Neo1 bound with PI4P, Neo1 in a final protein concentration of 1.058 mg ml$^{-1}$ was incubated with 10 mM NaF, 2 mM BeSO$_4$, 5 mM MgCl$_2$ and 0.5 mM PI4P dissolved in dimethyl sulfoxide/ethanol (v/v 1:1) at 4 °C overnight. After glow discharge cleaning for 15 s in a Gatan Solarus Model 950 plasma cleaner (Gatan), the 400-mesh gold C-Flat R1.2/1.3 holey carbon grids (Electron Microscopy Sciences) were applied with 3 μl of protein sample. A sample vitrification was carried out in a Vitrobot Mark IV (Thermo Fisher Scientific) at 5 °C and 100% relative humidity with the following settings: blot time: 4 s, blot force: 2 and wait time: 0 s. The EM grids were plunge-frozen in liquid ethane and cooled by liquid nitrogen. Automated cryo-EM data collection was performed on a 300 kV Titian Krios electron microscope controlled by SerialEM v4.0.12 (ref. 83) in multihole mode. The movie stacks were recorded at 105,000× magnification with the objective lens defocus values varied from −1.3 to −1.6 μm, in a K3 direct electron detector (Gatan) in super-resolution mode. The calibrated image pixel size was 0.828 Å at specimen level. A GIF Quantum energy filter was also used to remove inelastically scattered electrons with the slit width set to 20 eV. For each movie micrograph, a total of 50 frames were captured with 1 s exposure time and a total electron dose of 58 e$^-$ Å$^{-2}$.

## Cryo-EM image processing and 3D reconstruction
Image processing was carried out in cryoSPARC v4.2. The movie stacks were aligned by patch motion correction for beam-induced movement, then the contrast transfer function (CTF) parameter of each micrograph was determined by patch CTF estimation, and the CTF effect was subsequently corrected. After manual inspection to remove poor-quality images, the remaining micrographs were frame aligned and dose weighted. The raw particle images were first picked using templates generated from a preliminary three-dimensional (3D) map. The picked raw particles were subjected to a two-dimensional (2D) classification. After removing obvious 'junk' classes and also duplicates with a minimum separation distance of 50 Å, the resulting particles were used for ab initio 3D reconstructions. After three rounds of heterogeneous refinement, the good class particles were subjected to a 2D classification to select rare-view classes for Topaz[84]. The Topaz-picked raw particles were subjected to two rounds of heterogeneous refinement followed by a 2D classification, from which the good class particles were combined with previously template-picked good class particles. The combined and duplicate-removed particles were further processed by non-uniform refinement followed by local refinement to obtain the final map.

## Model building, refinement and validation
The initial atomic model was built by fitting the Neo1 AlphaFold model[85] into the EM map. The atomic model was then rebuilt manually and refined with Coot v0.9.8.3 (ref. 86) and ISOLDE v1.6 (ref. 87), followed by iterative real-space refinement in Phenix v1.20.1 (ref. 88). The final model was validated by MolProbity v4.5.1 (ref. 89). Structural figures were prepared in UCSF ChimeraX v1.7 (ref. 90).

## ATP hydrolysis assay
ATPase activity assays were conducted using BIOMOL Green Reagent (Enzo Life Sciences) to quantify released inorganic phosphate. Lipids PE and PI4P were solubilized with 20 mM and 4 mM sodium cholate, respectively, in 20 mM HEPES buffer, pH 7.4 and 150 mM NaCl. Each reaction mixture contained 0.025 mg ml$^{-1}$ protein, 0.0025% LMNG, 0.00025% CHS, 20 mM HEPES, pH 7.4, 150 mM NaCl, 10 mM MgCl$_2$ and 0.25 mM ATP. The reactions were carried out at 37 °C for 15 min and immediately terminated by the addition of the reagent. After incubation of the mixture for 20 min at room temperature, the absorbance at 620 nm was measured using a microplate reader SpectraMax M2e (Molecular Devices). The phosphate concentration was determined by calibration using the phosphate standard (BML-KI102).

## Statistics and reproducibility
All the data are from at least three independent biological replicates. No statistical method was used to predetermine sample size. The data distribution was assumed to be normal, but this was not formally tested. Data collection and analysis were not performed blind to the conditions of the experiments. No data were excluded from the analysis. The experiments were not randomized. All the growth assays, western blot, imaging assays, quantifications and statistics were derived from $n = 3$ independent experiments. All the yeast cell images were acquired on DeltaVision Elite Imaging System equipped with a 100×, 1.4 NA oil immersion objective lens followed by deconvolution using softWoRx software (v7.0.0; GE Healthcare). The mammalian cell images were acquired using Inverted LSM880-Airyscan (Zeiss) microscope equipped with a 63×, 1.4 NA oil immersion objective lens and deconvoluted by Zen software. The images were analysed and fluorescence intensity was quantified using ImageJ1.54g. The error bars represent the s.d. as indicated in the figure legends. The data were processed and analysed in GraphPad Prism 9.5.0. The statistical analysis was performed using an ordinary one-way or two-way analysis of variance (ANOVA) or mixed model analysis with multiple comparisons test as specified in the figure legends. The figures and graphs were assembled in Adobe Illustrator Version 27.0.

**Reporting summary**

Further information on research design is available in the Nature Portfolio Reporting Summary linked to this article.

## Data availability

The cryo-EM 3D map of the *S. cerevisiae* Neo1 in E2P state and bound with PI4P has been deposited in the Electron Microscopy Data Bank under accession code EMD-44850. The corresponding atomic model has been deposited in the Protein Data Bank (PDB) under accession code 9BS1. The EM data are available from the authors upon reasonable request. All other data supporting the findings of this study are available from the corresponding authors on reasonable request. Source data are provided with this paper.

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

## Acknowledgements

We thank the Graham lab members, R. O'Brien (Vanderbilt University) and J. York (Vanderbilt University) for helpful discussion during these studies and R. Zhang for performing neomycin sensitivity assays. We thank Y. Mao (Weill Institute for Cell and Molecular Biology) and C. Fromme (Weill Institute for Cell and Molecular Biology) for providing GFP–SidC$_{P4C}$ expression plasmids used in the studies. The cryo-EM images were collected in the David Van Andel Advanced Cryo-Electron Microscopy Suite at Van Andel Institute. We thank G. Zhao and X. Meng for facilitating data collection. This work was supported by the US National Institutes of Health (grant nos. R35GM144123 to T.R.G. and R01CA231466 to H.L.) and the Van Andel Institute (to H.L.). The materials generated in this study are freely available upon request to the corresponding authors. The experiments were performed in part through the use of the Vanderbilt Cell Imaging Shared Resource (supported by NIH grant no. 1 S10 OD021630 1).

## Author contributions

B.K.J., H.D.D., H.L. and T.R.G. conceived and designed the experiments. B.K.J. generated yeast Neo1 mutant expression constructs and performed all cellular studies. H.D.D. generated WT Neo1 expression construct, expressed and purified the WT and mutant proteins and carried out ATPase assays. H.D.D. performed cryo-EM and built the atomic model. C.V. and A.S. performed yeast growth assays. All authors analysed the data and edited the paper.

## Competing interests

The authors declare no competing interests.

## Additional information

**Extended data** are available for this paper at https://doi.org/10.1038/s41556-025-01692-z.

**Correspondence and requests for materials** should be addressed to Bhawik K. Jain or Todd R. Graham.

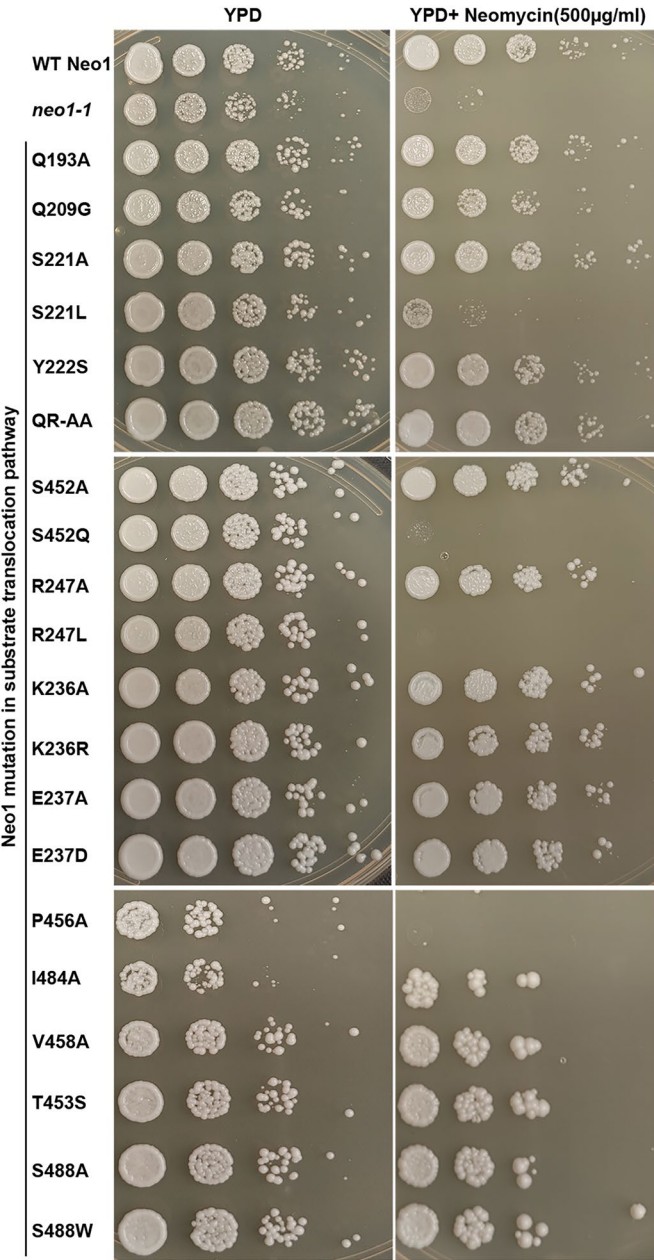

**Extended Data Fig. 1 | Neo1 substrate transport pathway mutants are sensitive to neomycin.** Neomycin sensitivity assay of *neo1* substrate transport pathway mutants at 26 °C on YPD and YPD Neomycin (500 µg/ml) plates.

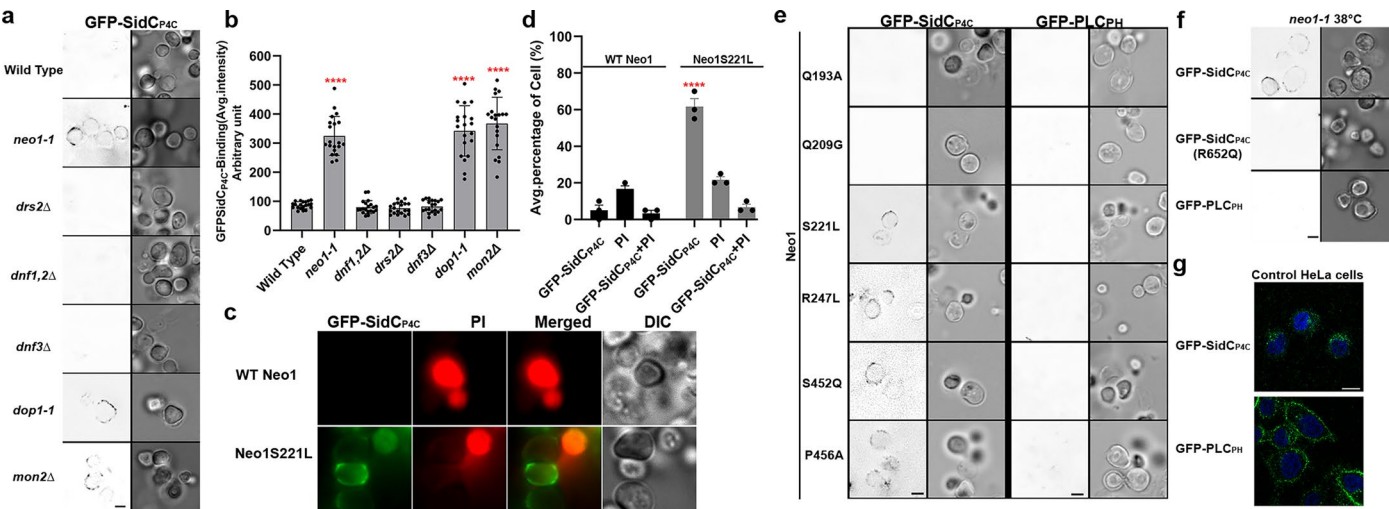

**Extended Data Fig. 2 | Neomycin sensitive *neo1* substrate transport pathway mutants expose PI4P in extracellular leaflets. a**, *neo1-1, dop1-1* and *mon2Δ* mutant exposes PI4P on extracellular leaflets probed using recombinantly purified GFP-SidC$_{P4C}$. The right panel is the fluorescence signal intensity of the GFP-probe and left panel shows the DIC panel to display yeast cells. Scale bar = 2 μm. **b**, Quantification of GFP-SidC$_{P4C}$ binding to the cell surface of the WT cells and flippase mutant strains. Two-way ANOVA was performed to test the variance and comparisons with WT cells were made with Tukey's multiple comparison test. (n = 20 cells from three biologically independent experiments, Data are mean ± (SD). **** P < 0.001). **c**, Intact live cells expose PI4P on the extracellular leaflet. Scale bar = 2 μm. **d**, Quantification of the percentage of the cells exposing PI4P stained with PI. Two-way ANOVA was performed to test the

variance and comparisons with WT Neo1 cells were made with Šídák's multiple comparisons test. (n = 3 biologically independent experiments, Data are mean ±(SD). **** P < 0.001). **e**, Perturbation of plasma membrane phosphoinositide asymmetry was tested in *neo1* mutant strains. (n = 20 cells from three biologically independent experiments). Scale bar = 2 μm. **f**, Only GFP-SidC$_{P4C}$ binds to neomycin-sensitive mutant cells at the rim of cells, suggesting exposure of PI4P on extracellular leaflet of the plasma membrane. (n = 20 cells from three biologically independent experiments). Scale bar = 2 μm. **g**, HeLa cells were fixed, permeabilized and stained with recombinant biosensors GFP-SidC$_{P4C}$ or GFP-PLC$_{PH}$. (n = 20 cells from three biologically independent experiments). Scale bar = 10 μm.

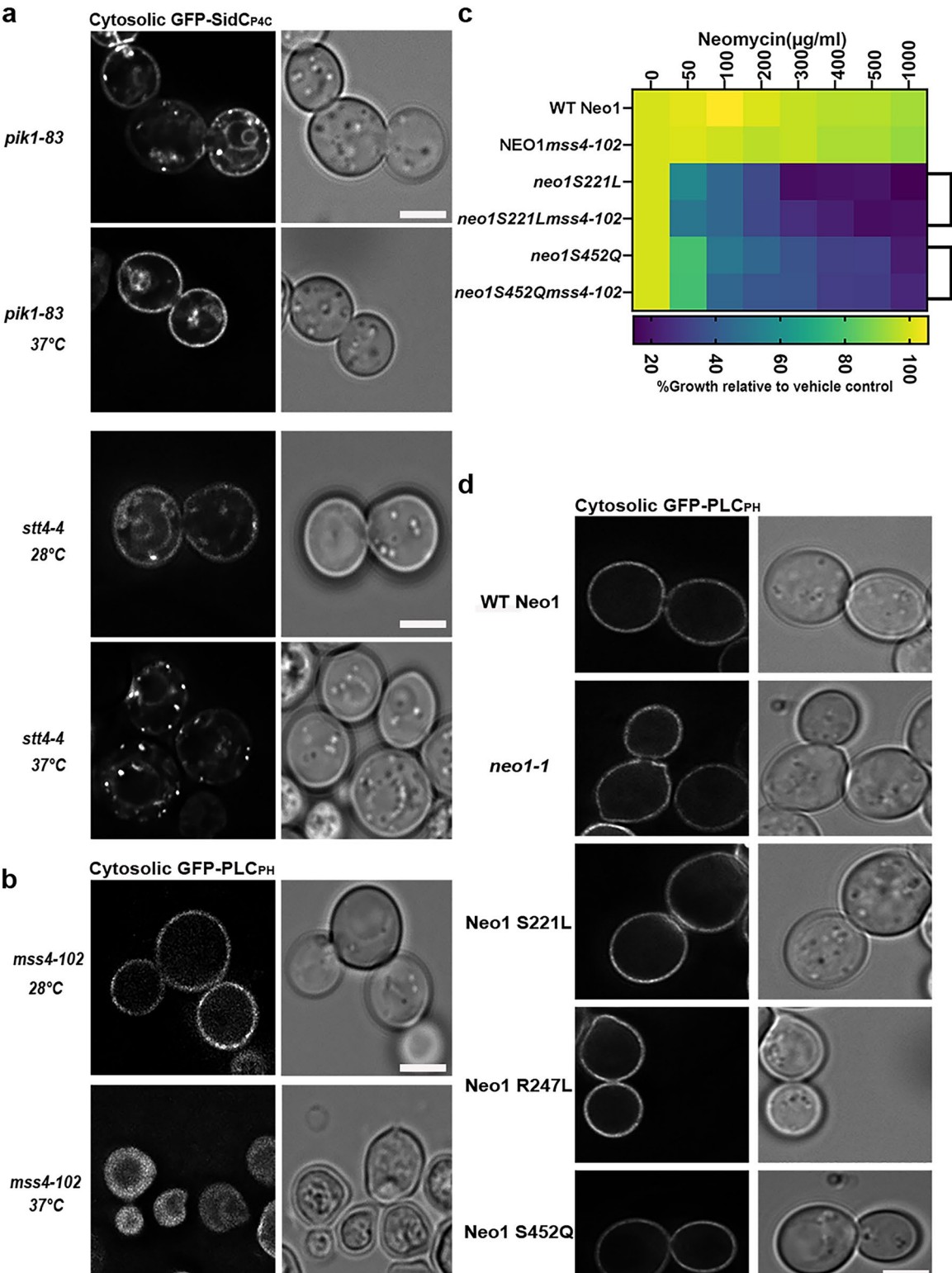

**Extended Data Fig. 3 | Confirmation of *pik1-83, stt4-4* and *mss4-102* mutants.**
**a**, Live cell fluorescence microscopy of intracellular (cytosolic) GFP-SidC$_{P4C}$
in PI kinase mutant cells *pik1-83* and *stt4-4*, at permissive (28 °C) and non-
permissive temperature (37 °C). (n = 20 cells from three biologically independent
experiments). **b**, Neo1 neomycin sensitive mutants do not have any effect on
localization of intracellular PI(4,5)P$_2$ probed with GFP-PLC$_{PH}$ expressed in the
yeast cytosol. Scale Bar =2 μm. n = 3 biological replicates.(n = 20 cells from three
biologically independent experiments). **c**, A PI 5-kinase Mss4 mutant allele fails

to suppress the neomycin sensitivity of *neo1* mutants. The data represent growth
relative to WT cells without the drug. Mixed model analysis was performed to
test the variance and comparisons with *neo1S221L* or *neo1S452Q* were made with
Tukey's multiple comparisons test (n = 3 biologically independent experiments,
Data are mean ± SD). ns represents not significant. **d**, localization of GFP-
PLC$_{PH}$ expressed in the cytosol of a *mss4-102* mutant at permissive (28 °C) and
non-permissive temperature (37 °C) Scale Bar = 2 μm.(n = 20 cells from three
biologically independent experiments).

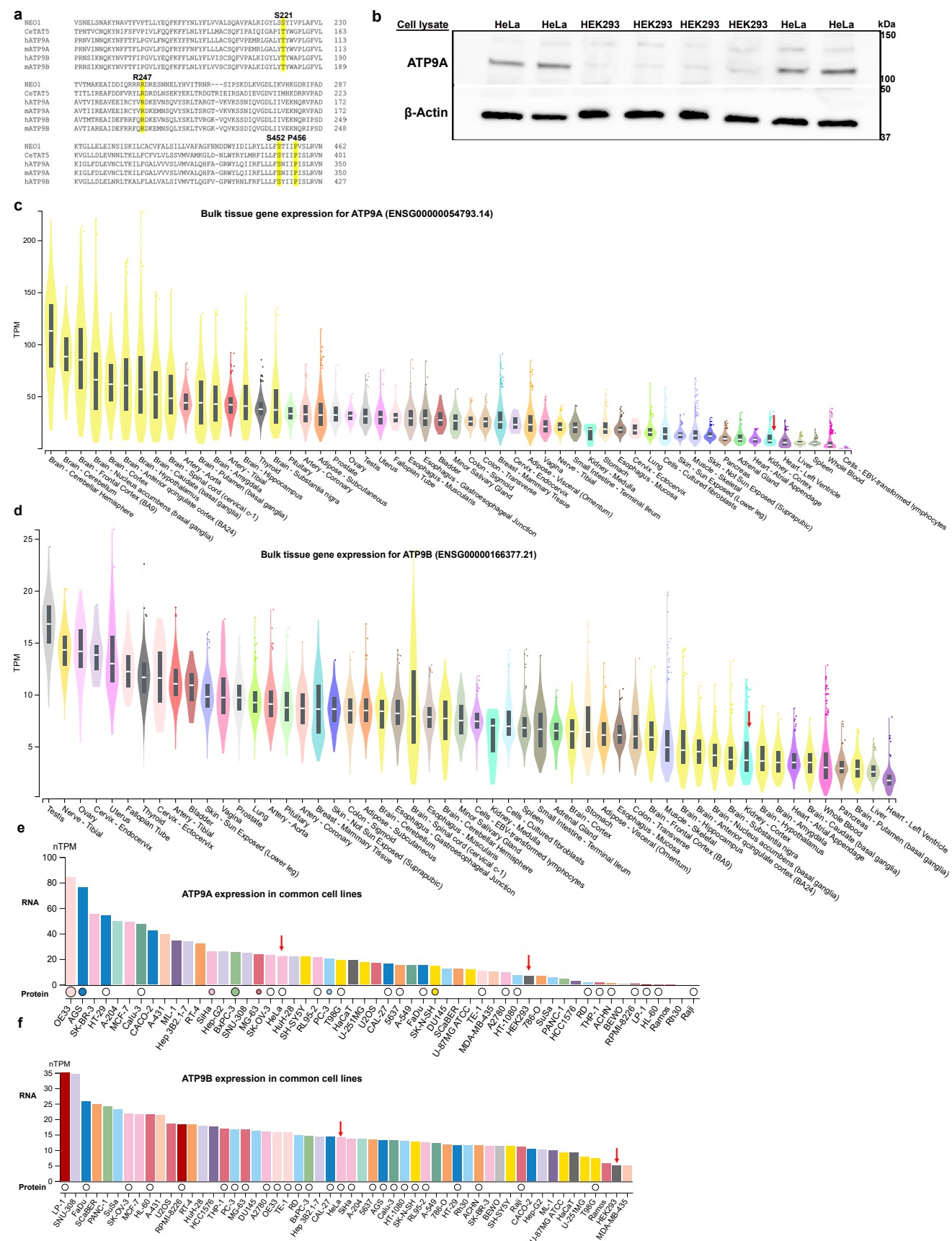

**Extended Data Fig. 4 | See next page for caption.**

**Extended Data Fig. 4 | Expression of ATP9A and ATP9B. a**, Sequence alignment of Neo1 with human and mouse orthologs ATP9A, ATP9B and *C. elegans* Tat5. **b**, Expression of ATP9A in HeLa cells and HEK293 cell lysates by immunoblotting. (n = 3 biologically independent experiments). **c,d** RNA-seq data of ATP9A and ATP9B in tissues. **e,f** RNA-seq data of ATP9A and ATP9B in commonly used cell lines. Arrow highlights the expression of ATP9A in Kidney tissues, HeLa cells

and HEK293 cells. The expression plot images were adopted from Genotype-Tissue Expression (GTEx) Portal and Human Protein Atlas (ATP9A: https://www.proteinatlas.org/ENSG00000054793-ATP9A, ATP9B: https://www.proteinatlas.org/ENSG00000166377-ATP9B). The data used for the analyses in c,d,e,f described in this manuscript were obtained from the GTEx Portal on 01/20/2025.

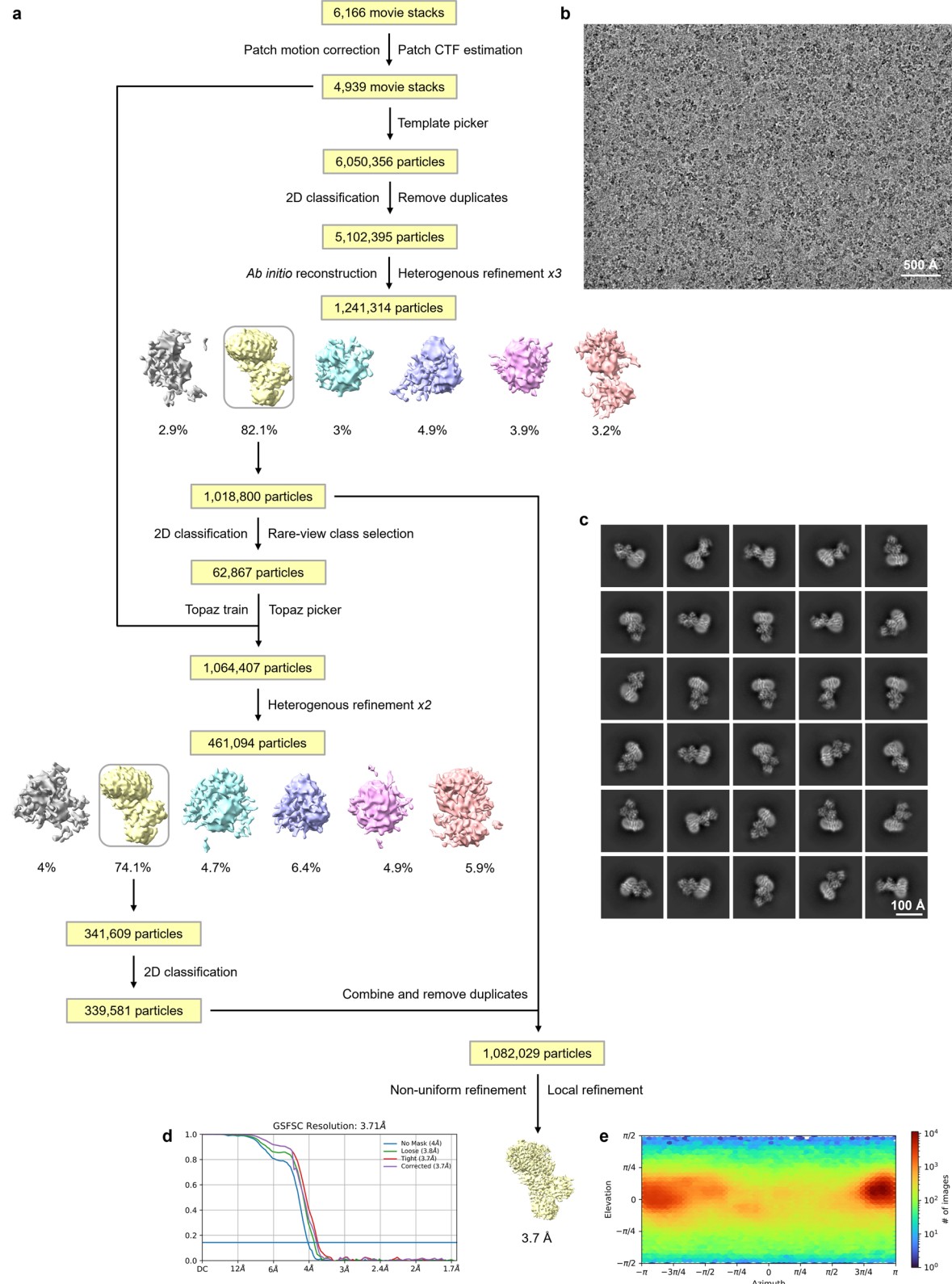

**Extended Data Fig. 5 | Cryo-EM structural determination of peptidisc-reconstituted Neo1 bound with PI4P in the E2P state. a**, Data processing workflow. **b**, Representative raw micrograph. A total of 6,166 such micrographs were recorded. **c**, Representative 2D classes. **d**, Gold-standard Fourier shell correlation (GSFSC) curve for the 3D reconstruction. **e**, Angular distribution heat map for the 3D reconstruction.

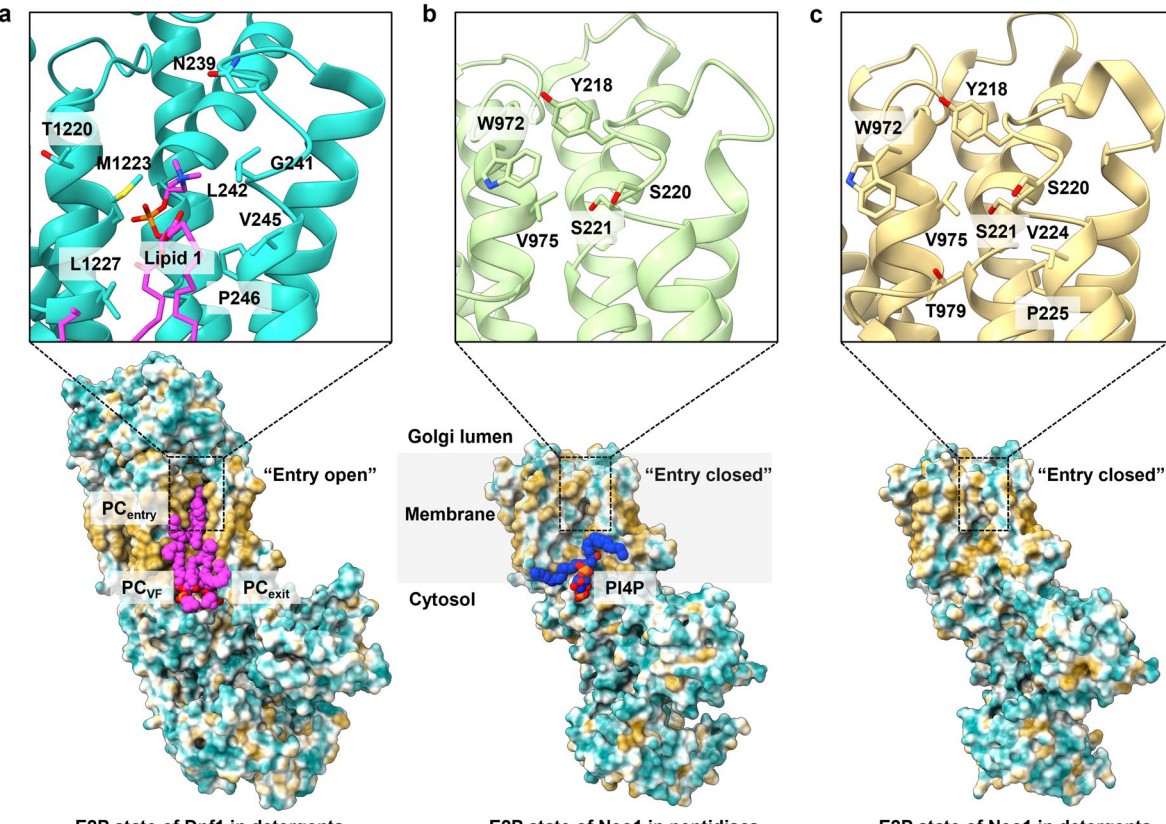

**E2P state of Dnf1 in detergents**

**E2P state of Neo1 in peptidiscs**

**E2P state of Neo1 in detergents**

**Extended Data Fig. 6 | Comparison of substrate binding sites in Neo1 and Dnf1.** Substrate entry site is open in the E2P state of Dnf1 in detergents (PDB ID 7KYC) (**a**), but is closed in the peptidisc-reconstituted Neo1 in E2P state (**b**, this study), and in the E2P state of Neo1 in detergents (PDB ID 7RD6) (**c**). Residues surrounding the substrate entry site are shown in sticks. The entry site is more tightly closed in detergent-reconstituted Neo1, involving additional residues T979, V224 and P225.

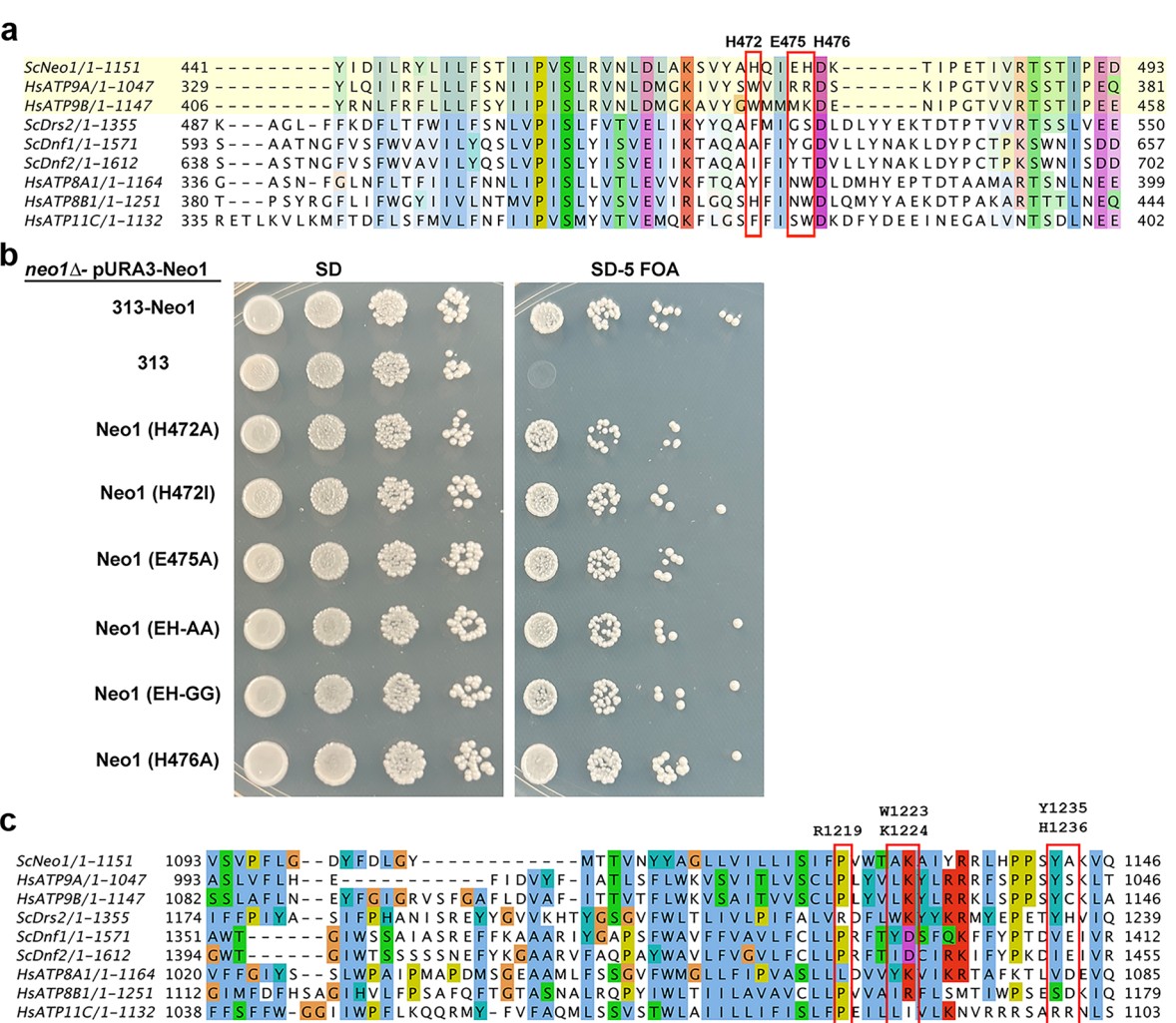

**Extended Data Fig. 7 | Mutation in the Neo1 PI4P binding region does not affect cell growth. a**, Sequence alignment of Neo1 PI4P binding region with human orthologs ATP9A, ATP9B and Drs2. *Sc*: *Saccharomyces cerevisiae*. *Hs*: *Homo sapiens*. **b**, All of the PI4P binding region Neo1 mutants support viability of *neo1Δ* cells. We transformed a *neo1Δ*pURA3-NEO1 strain with HIS3-marked plasmids harboring the indicated Neo1 variant. Cells were spotted on complete media

SD to select both plasmids and on SD-5-FOA plates to pop out pURA3-NEO1 plasmid which will allow the expression on 5-FOA plates. NEO1 is an essential gene, and all Neo1 mutants were able to complement the growth defect of *neo1Δ* strain. **c**, Sequence alignment of Drs2 PI4P binding region with Neo1, ATP9A and ATP9B. The Drs2 PI4P binding site residues are not well conserved in Neo1/ATP9A/ATP9B.

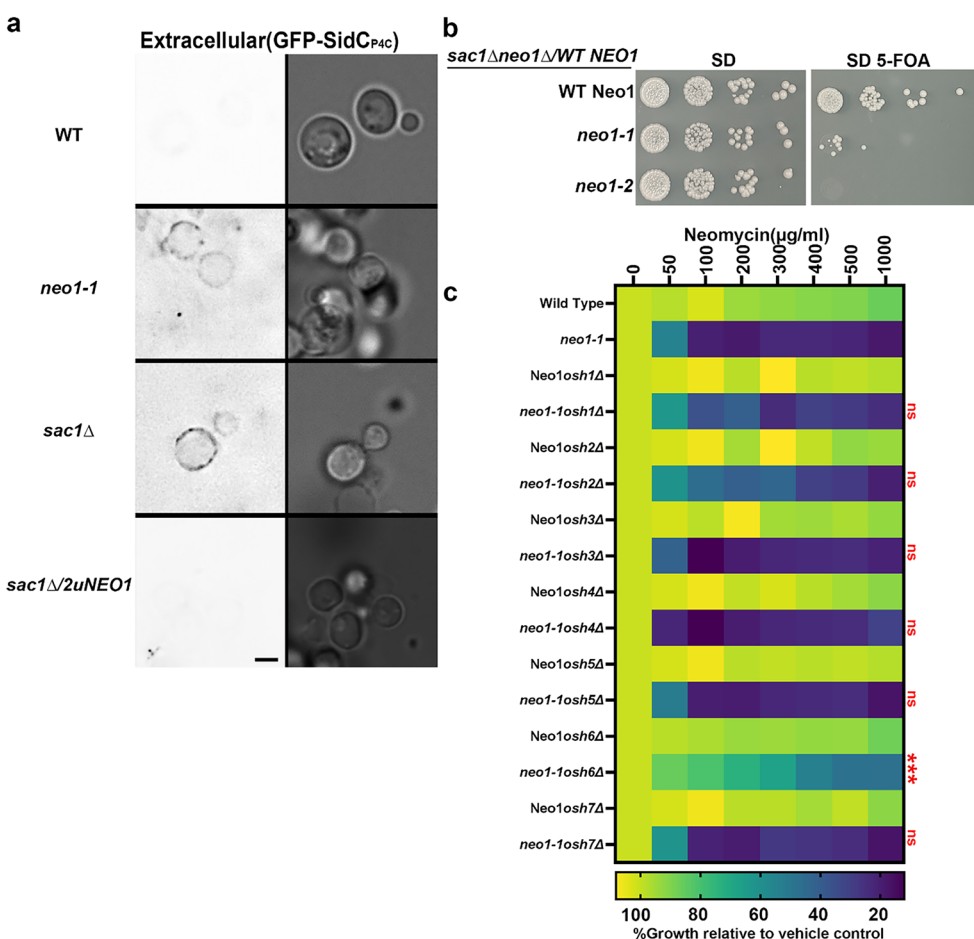

**Extended Data Fig. 8 | sac1Δ cells expose PI4P in extracellular leaflets and neomycin sensitivity assay of neo1-1 osh mutants. a**, sac1Δ cells expose PI4P in extracellular leaflets and overexpression of *NEO1* from a 2 μ plasmid suppresses the PI4P exposure. The right panel is the fluorescence signal intensity of the GFP-probe and left panel shows the DIC panel to display yeast cells. Scale Bar = 2 μm. n = 3 independent biological replicates. **b**, neo1 temperature sensitive mutants *(ts) neo1-1 sac1Δ or neo1-2 sac1Δ* mutants are inviable or grow poorly on 5-FOA plates. We transformed a *sac1Δ neo1Δ* pURA3-NEO1 strain with LEU-marked plasmids harboring the indicated Neo1 temperature sensitive variant. Cells were spotted on complete media SD to select both plasmids and on SD-5-FOA plates to pop out pURA3-NEO1 plasmid which will allow the expression of Neo1 variant on 5-FOA plates. **c**, Deletion of only *osh6Δ* suppresses the neomycin sensitivity of *neo1* mutants. The data represent growth relative to WT cells without the drug. Mixed model analysis was performed to test the variance and comparisons with *neo1-1* were made with Tukey's multiple comparison test (n = 3 biologically independent experiments, Data are mean ± SD). ns represents not significant. *** represents p value = 0.007).

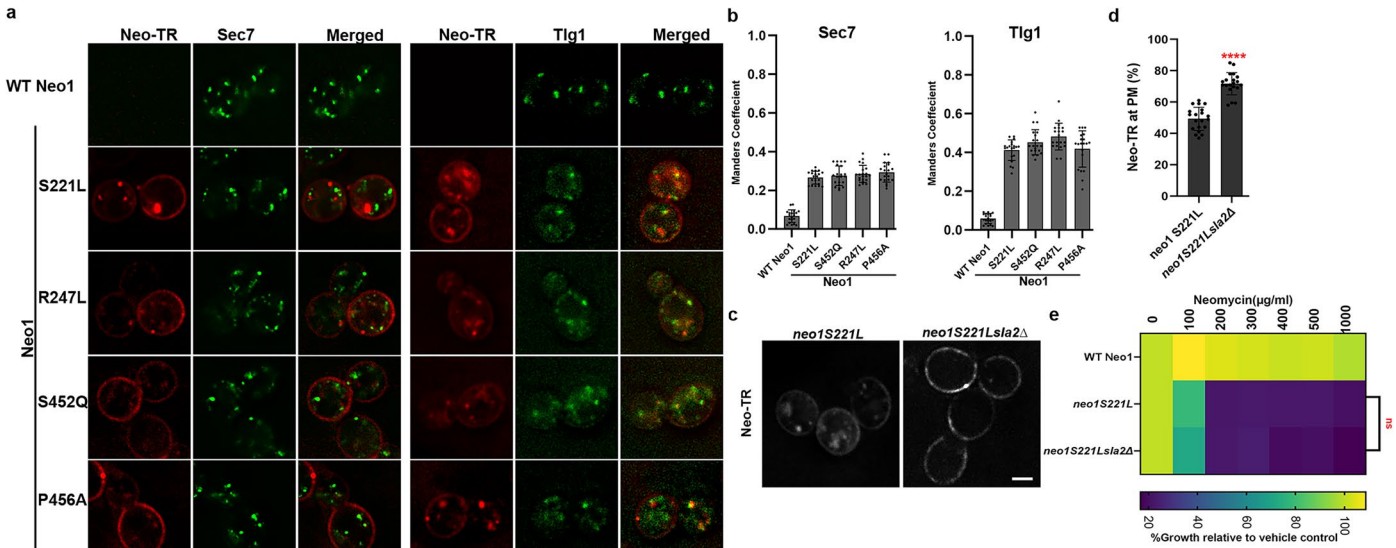

**Extended Data Fig. 9 | Neomycin binds to exposed PI4P and is endocytosed.**
**a**, Neo-TR binds neomycin-sensitive *neo1* mutants and endocytosed to the *trans*-Golgi Network (Sec7) and endosomal compartment (Tlg1). Scale bar = 2 μm. **b**, Colocalization of Neo-TR with TGN marker Sec7 and endosomal marker Tlg1 (n = 20 cells from three biologically independent experiments). Data are mean ± SD. **c**, knockout of *sla2Δ* in neomycin sensitive mutant *neo1 S221L* blocks Neo-TR internalization. Scale bar = 2 μm **d**, Quantification of Neo-TR fluorescence intensity at the PM. For all quantification, (*n* = 20 cells from three biologically independent experiments). The two-tailed unpaired t-test was used for comparison. Data are mean ±(SD). **** represents p < 0.0001. **e**, Deletion of *sla2Δ* fails to suppress the neomycin sensitivity of *neo1* mutants *neo1S221L*. The data represent growth relative to WT cells without the drug. Mixed model analysis was performed to test the variance and comparisons with *neo1S221L* were made with Tukey's multiple comparison test.(n = 3 three biologically independent experiments, Data are mean ± SD, ns represents not significant).

# Reporting Summary

## Statistics

For all statistical analyses, confirm that the following items are present in the figure legend, table legend, main text, or Methods section.

| n/a | Confirmed | |
|---|---|---|
| ☐ | ☒ | The exact sample size (*n*) for each experimental group/condition, given as a discrete number and unit of measurement |
| ☐ | ☒ | A statement on whether measurements were taken from distinct samples or whether the same sample was measured repeatedly |
| ☐ | ☒ | The statistical test(s) used AND whether they are one- or two-sided *Only common tests should be described solely by name; describe more complex techniques in the Methods section.* |
| ☐ | ☒ | A description of all covariates tested |
| ☒ | ☐ | A description of any assumptions or corrections, such as tests of normality and adjustment for multiple comparisons |
| ☐ | ☒ | A full description of the statistical parameters including central tendency (e.g. means) or other basic estimates (e.g. regression coefficient) AND variation (e.g. standard deviation) or associated estimates of uncertainty (e.g. confidence intervals) |
| ☐ | ☒ | For null hypothesis testing, the test statistic (e.g. *F*, *t*, *r*) with confidence intervals, effect sizes, degrees of freedom and *P* value noted *Give P values as exact values whenever suitable.* |
| ☒ | ☐ | For Bayesian analysis, information on the choice of priors and Markov chain Monte Carlo settings |
| ☒ | ☐ | For hierarchical and complex designs, identification of the appropriate level for tests and full reporting of outcomes |
| ☒ | ☐ | Estimates of effect sizes (e.g. Cohen's *d*, Pearson's *r*), indicating how they were calculated |

*Our web collection on statistics for biologists contains articles on many of the points above.*

## Software and code

Policy information about availability of computer code

| Data collection | Cryo-EM data were collected using SerialEM v4.0. Fluorescence images were acquired using a Delta Vision Elite Imaging System (GE Healthcare Life Sciences, Pittsburgh, PA) 100X,1.4 NA oil immersion objective lens. The images were deconvoluted by softWoRx software (v7.0.0; GE Healthcare) or Images were acquired using Inverted LSM880-Airyscan (Zeiss) microscope equipped with a 63×,1.4 NA oil immersion objective lens followed by deconvolution using Zen Desk software. |
|---|---|
| Data analysis | cryoSPARC v4.2.1, Coot v0.9.8.3, ISOLDE v1.6, Phenix v1.20.1, MolProbity v4.5.1, UCSF ChimeraX v1.7. Images were analyzed and fluorescence intensity was quantified using ImageJ1.54g. GraphPad Prism 9.5.0.730 |

For manuscripts utilizing custom algorithms or software that are central to the research but not yet described in published literature, software must be made available to editors and reviewers. We strongly encourage code deposition in a community repository (e.g. GitHub). See the Nature Portfolio guidelines for submitting code & software for further information.

## Data

Policy information about availability of data

All manuscripts must include a data availability statement. This statement should provide the following information, where applicable:
- Accession codes, unique identifiers, or web links for publicly available datasets
- A description of any restrictions on data availability
- For clinical datasets or third party data, please ensure that the statement adheres to our policy

The cryo-EM 3D map of the S. cerevisiae Neo1 in E2P state and bound with PI4P has been deposited in the Electron Microscopy Data Bank under accession code EMD-44850. The corresponding atomic model has been deposited in the Protein Data Bank under accession code 9BS1. Source data are provided with this paper. All other data supporting the findings of this study are available from the corresponding author on reasonable request.

## Research involving human participants, their data, or biological material

Policy information about studies with human participants or human data. See also policy information about sex, gender (identity/presentation), and sexual orientation and race, ethnicity and racism.

| | |
|---|---|
| Reporting on sex and gender | N/A |
| Reporting on race, ethnicity, or other socially relevant groupings | N/A |
| Population characteristics | N/A |
| Recruitment | N/A |
| Ethics oversight | N/A |

Note that full information on the approval of the study protocol must also be provided in the manuscript.

# Field-specific reporting

Please select the one below that is the best fit for your research. If you are not sure, read the appropriate sections before making your selection.

☒ Life sciences          ☐ Behavioural & social sciences          ☐ Ecological, evolutionary & environmental sciences

For a reference copy of the document with all sections, see nature.com/documents/nr-reporting-summary-flat.pdf

# Life sciences study design

All studies must disclose on these points even when the disclosure is negative.

| | |
|---|---|
| Sample size | The sample size for cryo-EM studies were determined by properties and qualities of the particles and also the number of particles available in each micrograph. For the current cryo-EM dataset, 6,166 raw micrographs were collected.<br>For cellular studies, No statistical methods was used to determined the sample sizes but our sample sizes are similar to those reported in the previous studies in the field. For all the cellular studies 3 independent biological isolates were used for the experiments. All the yeast growth assay, imaging were performed using 3 independent biological isolates. 3 Independent experiments were performed when siRNA studies were performed in the cell lines.<br>For fluorescence microscopy, No statistical methods was used to determined the sample sizes. Images from 20 cells were collected for 3 independent biological isolates or experiments. |
| Data exclusions | "Bad" raw particle images that did not produce 2D class averages or 3D class maps with defined features were excluded after 2D and 3D classifications. The criterion is empirical but is a standard image processing practice in the cryo-EM community. |
| Replication | Reproducibility resides in the large number of particles used to derive at the final 3D maps or 2D averages. The reliability and the resolution are measured by the Gold-standard Fourier shell correlation. Replication efforts with multiple refinement runs successfully yielded similar 3D maps. All the experiments were performed in the triplicates or 3 independent biological isolates were used. |
| Randomization | The allocation or selection of "good" and "bad" particles are determined by the computer program CryoSPARC based on the 2D templates or 3D volumes provided prior to running the program. |
| Blinding | The investigators cannot be blinded to the specific data points during data collection and analysis, because visual inspection is necessary to ascertain the data quality. There is no need for blinding in this type of study. |

# Reporting for specific materials, systems and methods

We require information from authors about some types of materials, experimental systems and methods used in many studies. Here, indicate whether each material, system or method listed is relevant to your study. If you are not sure if a list item applies to your research, read the appropriate section before selecting a response.

## Materials & experimental systems

| n/a | Involved in the study |
|---|---|
| ☐ | ☒ Antibodies |
| ☐ | ☒ Eukaryotic cell lines |
| ☒ | ☐ Palaeontology and archaeology |
| ☒ | ☐ Animals and other organisms |
| ☒ | ☐ Clinical data |
| ☒ | ☐ Dual use research of concern |
| ☒ | ☐ Plants |

## Methods

| n/a | Involved in the study |
|---|---|
| ☒ | ☐ ChIP-seq |
| ☒ | ☐ Flow cytometry |
| ☒ | ☐ MRI-based neuroimaging |

## Antibodies

| | |
|---|---|
| Antibodies used | Rabbit Anti-ATP9A antibody (ab234873, 1:1000) <br> Mouse Anti-β-Actin antibody ( CST#3700, 1:10000) <br> Anti-Mouse IgG (H+L), HRP conjugate (W4021, 1:10,000 dilution) Lot: 0000459067 Promega (Madison, WI). <br> Anti-Rabbit IgG (H+L), HRP conjugate (W4011, 1:10,000 dilution) Lot: 0000529943 Promega (Madison, WI). |
| Validation | Rabbit Anti-ATP9A antibody (ab234873, 1:1000) Used in (PMID-36715683) https://www.abcam.com/products/primary-antibodies/atp9a-antibody-ab234873.html <br> Mouse Anti-β-Actin antibody ( CST#3700, 1:10000)  https://www.cellsignal.com/products/primary-antibodies/b-actin-8h10d10-mouse-mab/3700 <br><br> Anti-Mouse IgG (H+L), HRP conjugate (W4021, 1:10,000) Lot: 0000459067 https://www.promega.com/products/protein-detection/primary-and-secondary-antibodies/anti_mouse-igg-h-and-l-hrp-conjugate/? catNum=W4021 <br> Anti-Rabbit IgG (H+L), HRP conjugate (W4011, 1:10,000) Lot: 0000529943 https ://www. pro mega. com/products/protein-detection/primary-an d-seco n da ry-a nti bodies/anti-rabbit -igg-h-a nd-1-h rp-co n jugate/? catNum=W4011 |

## Eukaryotic cell lines

Policy information about cell lines and Sex and Gender in Research

| | |
|---|---|
| Cell line source(s) | HeLa cells (RRID:CVCL_0030), HEK293 cells (ATCC, RRID:CVCL_1573 ) . |
| Authentication | Cell lines obtained were not authenticated. |
| Mycoplasma contamination | Cell lines were not tested for mycoplasma contamination. |
| Commonly misidentified lines (See ICLAC register) | No misidentified cell lines were used in the study |

## Plants

| | |
|---|---|
| Seed stocks | *Report on the source of all seed stocks or other plant material used. If applicable, state the seed stock centre and catalogue number. If plant specimens were collected from the field, describe the collection location, date and sampling procedures.* |
| Novel plant genotypes | *Describe the methods by which all novel plant genotypes were produced. This includes those generated by transgenic approaches, gene editing, chemical/radiation-based mutagenesis and hybridization. For transgenic lines, describe the transformation method, the number of independent lines analyzed and the generation upon which experiments were performed. For gene-edited lines, describe the editor used, the endogenous sequence targeted for editing, the targeting guide RNA sequence (if applicable) and how the editor was applied.* |
| Authentication | *Describe any authentication procedures for each seed stock used or novel genotype generated. Describe any experiments used to assess the effect of a mutation and, where applicable, how potential secondary effects (e.g. second site T-DNA insertions, mosiacism, off-target gene editing) were examined.* |

