## [Peer Review File · Nature Cell Biology]

P4-ATPases control phosphoinositide membrane asymmetry and neomycin resistance

Corresponding Author: Professor Todd Graham

Version 0:

Decision Letter:

*Please delete the link to your author homepage if you wish to forward this email to co-authors.

Dear Professor Graham,

Thank you again for submitting your manuscript, "P4-ATPase control over phosphoinositide membrane asymmetry and neomycin resistance", to Nature Cell Biology. It has now been seen by 3 referees, who are experts in phosphoinositides, vesicular transfer (Referee #1); cryoEM, transporters (Referee #2); and lipid scrambling (Referee #3). As you will see from their comments (attached below), they found the work of potential interest but have raised substantial concerns, which in our view would need to be addressed with considerable revisions before we can consider publication in Nature Cell Biology.

Nature Cell Biology editors discuss the referee reports in detail within the editorial team, including the chief editor, to identify key referee points that should be addressed with priority, and requests that are overruled as being beyond the scope of the current study. To guide the scope of the revisions, I have listed these points below. Our standard revision period is six months, and we are committed to providing a fair and constructive peer-review process, so please feel free to contact me if you would like to discuss any of the referee comments further.

In particular, it would be essential to:

1- The reviewers had questions about neomycin sensitivity that should be clarified and addressed:

Rev#3 points #1, #9

2- Further work is also needed to support and explore the link between PI4P and ATP hydrolysis by Neo1 as well as Neo1 regulation:

Rev#2 points #5-6-7

Rev#3 points #2-3

3- We recommend strengthening the endocytosis studies as per Rev#3's point #8 and addressing all other technical points and other referee concerns pertaining to strengthening existing data, providing controls, methodological details, clarifications and textual changes.

4- Finally, please pay close attention to our guidelines on statistical and methodological reporting (listed below) as failure to do so may delay the reconsideration of the revised manuscript. In particular, please provide:

We would be happy to consider a revised manuscript that would satisfactorily address these points, unless a similar paper is published elsewhere, or is accepted for publication in Nature Cell Biology in the meantime.

- ensure that it conforms to our format instructions and publication policies (see below and <https://www.nature.com/nature/for->

authors).

- provide a point-by-point rebuttal to the full referee reports verbatim, as provided at the end of this letter.

- provide the completed Reporting Summary (found here <https://www.nature.com/documents/nr-reporting-summary.pdf>). This is essential for reconsideration of the manuscript will be available to editors and referees in the event of peer review. For more information see <http://www.nature.com/authors/policies/availability.html> or contact me.

Nature Cell Biology is committed to improving transparency in authorship. As part of our efforts in this direction, we are now requesting that all authors identified as 'corresponding author' on published papers create and link their Open Researcher and Contributor Identifier (ORCID) with their account on the Manuscript Tracking System (MTS), prior to acceptance. ORCID helps the scientific community achieve unambiguous attribution of all scholarly contributions. You can create and link your ORCID from the home page of the MTS by clicking on 'Modify my Springer Nature account'. For more information please visit www.springernature.com/orcid.

This journal strongly supports public availability of data. Please place the data used in your paper into a public data repository, or alternatively, present the data as Supplementary Information. If data can only be shared on request, please explain why in your Data Availability Statement, and also in the correspondence with your editor. Please note that for some data types, deposition in a public repository is mandatory - more information on our data deposition policies and available repositories appears below.

Link Redacted

We hope that you will find our referees' comments and editorial guidance helpful. Please do not hesitate to contact me if there is anything you would like to discuss. Thank you again for considering NCB for your work,

Best wishes,

Melina

Melina Casadio, PhD
Senior Editor, Nature Cell Biology
ORCID ID: <https://orcid.org/0000-0003-2389-2243>

Reviewers' Comments:

Reviewer #1:

Remarks to the Author:

The manuscript resolves a couple of long-standing questions in cell biology. Neomycin is known to bind PIPs on the exofacial leaflet of cells. This study demonstrates that PI4P can be exposed on the surface of cells following a multi-step process requiring transfer to the ER, scrambling to the lumen, and transport via the secretory pathway back to the surface. The study also reveals that the flippase Neo1 can use PI4P as a substrate and translocate PI4P to the cytosolic leaflet, thereby limiting its exposure to the cell surface. Historically, P4-flippases have always been considered as aminophospholipid transferase only using PC, PE and PS as substrates. This is the first demonstration that they can also translocate a phosphoinositide. Taken together these observations explain why neo1 mutants are hyper-sensitive to neomycin.

The manuscript uses a combination of cell growth, microscopy and Cryo-EM techniques. Numerous point mutants are also examined to support the conclusions. The study also examines many yeast strains containing multiple double mutants to define the movement of PI4P. Statistics are appropriate for this work.

Based on the data, I concur with the authors' interpretations. This is an excellent and intriguing study.

I do have a few comments but I don't believe any new experiments are essential.

I wonder if an engineered Sac1 expressed on the cell surface protects neo1 cells from neomycin. Does Neo1's flipping of PI4P in the Golgi support secretion or transport to the vacuole?

These would likely be rather incremental additions to the current study. Unless the other reviewer(s) deem them essential, I don't believe they are worth delaying this manuscript. If the authors have tried these experiments or have insights perhaps they can add a comment or two to the text.

Figure 3 is thin on results. I suggest adding some of the extended data figures to this. Perhaps Extended Fig 5.

There may be a missing extended data figure panels, or the legend is wrong. "Extended Data Fig. 3: Confirmation of ATP9A knockdown and pik1-83, stt4-4 and mss4-102 mutants. a, Sequence alignment of Neo1 with human and mouse orthologs ATP9A, ATP9B and C. elegans Tat5...." doesn't apply to figure.

Figure 5b. The TR-neomycin is seen on the PM, endosomes and vacuole as the authors claim. However, in some of the fields, it looks like there are dead cells with very intense PM-only TR signals. I suggest being more selective in the choice of images. If the bright dead cells are eliminated, the images could be contrasted more without saturation. All of the panels have the same pixel size; however, zooming in a bit more on the relevant individual cells would also make for clearer images. In this case, each individual panel would need its own scale bar.

Additional contrasting would benefit Figures 1b and 5d. They are quite difficult to see on my monitors.

The finding that Osh6 is the predominant supplier of PI4P is a bit surprising. Does over-expression of OSH7 produce a similar effect? or over-expression of OSH4. It would suggest that either Osh6 functions at a much higher rate or possibly that Sac1 more efficiently dephosphorylates PI4P transferred by Osh4. Perhaps the difference is between peripheral ER and nuclear ER. I don't believe this requires direct experimentation for this manuscript, but perhaps a sentence could be added to the text, raising the possibilities.

Congrats on a fantastic study
Greg Fairn

Reviewer #2:

Remarks to the Author:

This work reveals that Neo1 flips PI4P across the Golgi membrane to the cytosolic leaflet, preventing PI4P exposure and thereby conferring neomycin resistance. Using multiple technologies, including cellular, biochemical, and structural studies, the authors demonstrated that PI4P is indeed the substrate of Neo1. The loss of Neo1 function results in the secretion of extracellular PI4P, which acts as a neomycin receptor and increases susceptibility.

Overall, this work provides extensive evidence supporting the results, and I believe it is suitable for publication in NCB. However, a few issues need to be addressed before final acceptance:

1. Line 87: The term "dnf" needs to be clearly described.
2. Lines 102-103: Please explain what "P4C" and "PH" stand for.
3. Line 113: Why was only S221L selected for the assay among the subsets of mutants? Please explain the reasons.
4. Lines 154-156: In Fig. 2h, the stimulation level by PS is similar to that of the combination of PS and PI4P in the 4th bar. Please provide an explanation for this phenomenon.
5. Fig 3: An overall structure of Neo1 is needed, and the RMSD of the structural superposition should be indicated. Additionally, the structure of Neo1 should be recolored to better distinguish it from Drs2.
6. Lines 174-176: Are the residues coordinating the PI4P headgroup in Drs2 also conserved in Neo1? If so, will mutations of these residues impair the function, such as the PI4P-stimulated ATPase activity of Neo1? It is possible that PI4P functions both as a regulator and a substrate.
7. The subsets of mutants S211L, R247L, S452Q, and P456A should be mapped in the structure to better interpret how they affect the function of Neo1. Additionally, the PI4P-stimulated ATPase activity of these mutants needs to be assessed.

Reviewer #3:

Remarks to the Author:

In this manuscript, Jain and colleagues demonstrated that the loss of Neo1 results in the exposure of PI4P on plasma membranes. While Neo1 was previously shown to possess flippase activities toward phosphatidylserine (PS) and phosphatidylethanolamine (PE), this study identifies PI4P as a new substrate of Neo1. Indeed, Cryo-EM analysis revealed

that PI4P is embedded in the substrate translocation pathway. The disruption of Neo1 activity in the Golgi apparatus altered the distribution of PI4P in the plasma membrane. A potential mechanism involves the disruption of the asymmetrical distribution of PI4P at the endoplasmic reticulum (ER), possibly by an unidentified scramblase, leading to the mis-localization of PI4P, which is then transferred to the plasma membrane through the Golgi apparatus. This exposed PI4P increases the sensitivity of eukaryotic cells to neomycin treatment. The mechanisms underlying PI4P exposure at the plasma membrane are intriguing, but the manuscript requires further improvement, especially about neomycin sensitivity, before publication.

1. It is known that only some types of eukaryotic cells show neomycin sensitivity. Do those cell types display PI4P on the cell surface? Can the expression level of ATP9A/B explain the sensitivity of such cells? Is there any correlation between the level of ATP9A/B and sensitivity to neomycin among different cell types?
2. In Fig. 2i, biochemical analysis needs to be performed with different concentrations of substrates to correctly understand the enzyme and substrate relationship.
3. In Fig. 2i, biochemical analysis needs to be done with a neomycin-sensitive mutant of NEO1 to clarify the enzyme and substrate relationship, especially for PI4P and PS or PE.
4. In Extended Data Fig. 2g, the data does not convincingly support the probe specificity. If the authors want to demonstrate probe specificity, it would be better to perform genetic manipulations to increase/decrease PI4P/PI(4,5)P2 expression and show corresponding changes in signal intensity.
5. In Extended Data Figs. 2a, 2e, 2f, and 6b, the background signal varies significantly. Some data appears overly white, possibly due to excessive contrast manipulation. Can the authors clarify this point? In addition, what accounts for the differences between the left and right panels?
6. In Extended Data Fig. 3, the legend description does not match the actual figure.
7. In Figs. 5b and 5c, it would be better to demonstrate localization to the Golgi and endosome by co-staining these organelles with markers.
8. In Fig. 5d, is latrunculin A treatment appropriate for endocytosis inhibition? Targeting actins affects functions other than endocytosis, so it might not necessarily prove the role of endocytosis. Is it possible to use other drugs such as Filipin, CPM, Dynasore, or Dyngo-5, or to perform genetic disruption of endocytosis machinery?
9. In Fig. 5, can the Neo-TR assay be conducted in mammalian cells such as HeLa cells? This data would demonstrate that neomycin binding to PI4P is conserved in mammalian cells. Similarly to question 1, are neomycin-susceptible cells labeled with Neo-TR more?
10. The role of Osh6 should be described in the manuscript. Why could Osh6 Δ rescue the neo1-I phenotype?
11. Can the authors show that PI4P exists on both the luminal and cytoplasmic sides of ER? This data supports the current hypothesis.
12. In Figs. 2i, 5c, and 5e, statistical comparison is lacking.
13. In Fig. 4a, the full names of each abbreviation need to be provided in the figure legend.

Minor Points

1. In the abstract, it is better to mention earlier that PI4P serves as a neomycin receptor.
2. Line 42: references 2 and 5 should be superscripted.
3. Line 155: should refer to Fig. 2i, not 2h.
4. Lines 165 and 177: Fig. 3, there are no sub-panels.
5. Line 229: should refer to Fig. 5a since there is no other sentence referring to that panel.
6. In Extended Data Fig. 6a, more explanation is needed (what is neo1-2? what is 5-FOA?).
7. References: some names are capitalized (e.g., references 2 and 5).
8. In Extended Data Fig. 2, PI doesn't seem to be staining the nucleus.

Methods should be written concisely, but should contain all elements necessary to allow interpretation and replication of the results. As a guideline, Methods sections typically do not exceed 3,000 words. The Methods should be divided into subsections listing reagents and techniques. When citing previous methods, accurate references should be provided and any alterations should be noted. Information must be provided about: antibody dilutions, company names, catalogue numbers and clone numbers for monoclonal antibodies; sequences of RNAi and cDNA probes/primers or company names and catalogue numbers if reagents are commercial; cell line names, sources and information on cell line identity and authentication. Animal studies and experiments involving human subjects must be reported in detail, identifying the committees approving the protocols. For studies involving human subjects/samples, a statement must be included confirming that informed consent was obtained. Statistical analyses and information on the reproducibility of experimental results should be provided in a section titled "Statistics and Reproducibility".

All Nature Cell Biology manuscripts submitted on or after March 21 2016 must include a Data availability statement as a separate section after Methods but before references, under the heading "Data Availability". For Springer Nature policies on data availability see <http://www.nature.com/authors/policies/availability.html>; for more information on this particular policy see <http://www.nature.com/authors/policies/data/data-availability-statements-data-citations.pdf>. The Data availability statement should include:

- Accession codes for primary datasets (generated during the study under consideration and designated as "primary accessions") and secondary datasets (published datasets reanalysed during the study under consideration, designated as "referenced accessions"). For primary accessions data should be made public to coincide with publication of the manuscript. A list of data types for which submission to community-endorsed public repositories is mandated (including sequence, structure, microarray, deep sequencing data) can be found here <http://www.nature.com/authors/policies/availability.html#data>.
- Unique identifiers (accession codes, DOIs or other unique persistent identifier) and hyperlinks for datasets deposited in an approved repository, but for which data deposition is not mandated (see here for details <http://www.nature.com/sdata/data-policies/repositories>).
- At a minimum, please include a statement confirming that all relevant data are available from the authors, and/or are included with the manuscript (e.g. as source data or supplementary information), listing which data are included (e.g. by figure panels and data types) and mentioning any restrictions on availability.

- If a dataset has a Digital Object Identifier (DOI) as its unique identifier, we strongly encourage including this in the Reference list and citing the dataset in the Methods.

We recommend that you upload the step-by-step protocols used in this manuscript to protocols.io. More details can found at <https://www.protocols.io/help/publish-articles>.

All imaging data should be accompanied by scale bars, which should be defined in the legend.

Cropped images of gels/blots are acceptable, but need to be accompanied by size markers, and to retain visible background signal within the linear range (i.e. should not be saturated). The boundaries of panels with low background have to be demarked with black lines. Splicing of panels should only be considered if unavoidable, and must be clearly marked on the figure, and noted in the legend with a statement on whether the samples were obtained and processed simultaneously. Quantitative comparisons between samples on different gels/blots are discouraged; if this is unavoidable, it should only be performed for samples derived from the same experiment with gels/blots were processed in parallel, which needs to be stated in the legend.

The total number of Supplementary Figures (not including the “unprocessed scans” Supplementary Figure) should not exceed the number of main display items (figures and/or tables (see our Guide to Authors and March 2012 editorial <http://www.nature.com/ncb/authors/submit/index.html#suppinfo>; <http://www.nature.com/ncb/journal/v14/n3/index.html#ed>). No restrictions apply to Supplementary Tables or Videos, but we advise authors to be selective in including supplemental data.

GUIDELINES FOR EXPERIMENTAL AND STATISTICAL REPORTING

REPORTING REQUIREMENTS – We are trying to improve the quality of methods and statistics reporting in our papers. To that end, we are now asking authors to complete a reporting summary that collects information on experimental design and reagents. The Reporting Summary can be found here <https://www.nature.com/documents/nr-reporting-summary.pdf> If you would like to reference the guidance text as you complete the template, please access these flattened versions at <http://www.nature.com/authors/policies/availability.html>.

We strongly recommend the presentation of source data for graphical and statistical analyses as a separate Supplementary Table, and request that source data for all independent repeats are provided when representative experiments of multiple independent repeats, or averages of two independent experiments are presented. This supplementary table should be in Excel format, with data for different figures provided as different sheets within a single Excel file. It should be labelled and numbered as one of the supplementary tables, titled “Statistics Source Data”, and mentioned in all relevant figure legends.

Version 1:

Decision Letter:

Our ref: NCB-A54828A

1st April 2025

Dear Dr. Graham,

Thank you for submitting your revised manuscript "P4-ATPase control over phosphoinositide membrane asymmetry and neomycin resistance" (NCB-A54828A). It has now been seen by the original referees and their comments are below. The reviewers find that the paper has improved in revision, and therefore we'll be happy in principle to publish it in Nature Cell Biology, pending minor revisions to satisfy the referees' final requests and to comply with our editorial and formatting guidelines.

Thank you again for your interest in Nature Cell Biology Please do not hesitate to contact me if you have any questions.

Sincerely,

Angela R Parrish, PhD
Locum Senior Editor
Nature Cell Biology

Reviewer #1 (Remarks to the Author):

The authors have addressed by minor comments. The additional experiments to address the comments of Reviewers 2 and 3 only further strengthen an outstanding study. Congratulations on the elegant work.

Regards, Greg Fairn

Reviewer #2 (Remarks to the Author):

I have no further comments. I recommend this paper to be accepted.

Reviewer #3 (Remarks to the Author):

The authors have effectively addressed all the points I raised, making the article clearer and more compelling for a broad audience. I appreciate their efforts and congratulate them on these exciting findings. I now strongly support its publication in Nature Cell Biology.

Version 2:

Decision Letter:

Dear Dr Graham,

I am pleased to inform you that your manuscript, "P4-ATPases control phosphoinositide membrane asymmetry and neomycin resistance", has now been accepted for publication in Nature Cell Biology. Congratulations!

Please note that *Nature Cell Biology* is a Transformative Journal (TJ). Authors may publish their research with us through the traditional subscription access route or make their paper immediately open access through payment of an article-processing charge (APC). Authors will not be required to make a final decision about access to their article until it has been accepted. <https://www.springernature.com/gp/open-research/transformative-journals> Find out more about Transformative Journals

Authors may need to take specific actions to achieve [compliance with funder and institutional open access mandates](https://www.springernature.com/gp/open-research/funding/policy-compliance-faqs). If your research is supported by a funder that requires immediate open access (e.g. according to [Plan S principles](https://www.springernature.com/gp/open-research/plan-s-compliance)) then you should select the gold OA route, and we will direct you to the compliant route where possible. For authors selecting the subscription publication route, the journal's standard licensing terms will need to be accepted, including [self-archiving policies](https://www.springernature.com/gp/open-research/policies/journal-policies). Those licensing terms will supersede any other terms that the author or any third party may assert apply to any version of the manuscript.

If you have not already done so, we strongly recommend that you upload the step-by-step protocols used in this manuscript to protocols.io (<https://protocols.io>), an open online resource that allows researchers to share their detailed experimental know-how. All uploaded protocols are made freely available and are assigned DOIs for ease of citation. Protocols and Nature Portfolio journal papers in which they are used can be linked to one another, and this link is clearly and prominently visible in the online versions of both. Authors who performed the specific experiments can act as primary authors for the Protocol as they will be best placed to share the methodology details, but the Corresponding Author of the present research paper should be included as one of the authors. By uploading your Protocols onto protocols.io, you are enabling researchers to more readily reproduce or adapt the methodology you use, as well as increasing the visibility of your protocols and papers. You can also establish a dedicated workspace to collect your lab Protocols. Further information can be found at <https://www.protocols.io/help/publish-articles>.

Nature Cell Biology encourages authors presenting evidence for cell, biological, molecular, and genetic interactions to consider communicating these findings using Biofactoid (<https://biofactoid.org/>). This tool helps users share a searchable representation of interactions (e.g. binding, gene expression, post-translational modification) between genes, gene products, or chemicals. Information added to Biofactoid, with author attribution, is shared on social media and public databases, such

as Pathway Commons, where it can be discovered and analyzed in the context of a large and growing corpus of knowledge.

With kind regards,

Angela R Parrish, PhD
Locum Senior Editor
Nature Cell Biology

** Visit the Springer Nature Editorial and Publishing website at http://editorial-jobs.springernature.com?utm_source=ejp_NCB_email&utm_medium=ejp_NCB_email&utm_campaign=ejp_NCB for more information about our career opportunities. If you have any questions please click [here](mailto:editorial.publishing.jobs@springernature.com).

Todd R. Graham

Stevenson Chair of Biological Sciences
Professor of Cell and Developmental Biology
Vanderbilt University

Point-by-point response to Reviewers

Reviewer 1:

1. I wonder if an engineered Sac1 expressed on the cell surface protects neo1 cells from neomycin. Does Neo1's flipping of PI4P in the Golgi support secretion or transport to the vacuole? These would likely be rather incremental additions to the current study. Unless the other reviewer(s) deem them essential, I don't believe they are worth delaying this manuscript. If the authors have tried these experiments or have insights perhaps they can add a comment or two to the text.

We thank the reviewer for this excellent suggestion. We expressed and purified recombinant Sac1 to test its ability to suppress neomycin sensitivity in human cells. Both control HEK293 cells and ATP9A knockdown cells exhibit sensitivity to neomycin. These cells were incubated with Sac1 and neomycin for 48 hours. Our data indicate that Sac1 treatment reduces neomycin sensitivity in both HEK293 and ATP9A knockdown cells. We have not included this result in the current manuscript because there are clinical implications for these data, and we want to explore this phenomenon in much more detail for another manuscript. Engineering Sac1 to flip its topology is a great idea, and we are working on doing these experiments, but hopefully for the next manuscript.

Previously, we observed that *neo1ts* mutants or depletion of Neo1p result in defects in protein transport, vacuole morphology, and vacuolar pH (Hua et al., 2003). We are using the collection of separation-of-function *neo1* mutants to examine these phenotypes. The results are fascinating and thus far suggest that PS transport by Neo1 is important for COPI function at the early Golgi, PI4P transport is critical for Snc1 (SNARE) recycling (probably exocytosis), and PE

transport is important for vacuole fusion and pH regulation. We already have a substantial amount of data for this project and we hope the reviewer will give us the latitude to publish these observations in a different manuscript.

2. Figure 3 is thin on results. I suggest adding some of the extended data figures to this. Perhaps Extended Fig 5.

We agree and have revised the structural figure and included more data in the new Figure 4.

3. There may be a missing extended data figure panels, or the legend is wrong. "Extended Data Fig. 3: Confirmation of ATP9A knockdown and pik1-83, stt4-4 and mss4-102 mutants. a, Sequence alignment of Neo1 with human and mouse orthologs ATP9A, ATP9B and C. elegans Tat5...." doesn't apply to figure.

We thank the reviewer for catching the mistake. We have fixed this in the revised manuscript.

4. Figure 5b. The TR-neomycin is seen on the PM, endosomes and vacuole as the authors claim. However, in some of the fields, it looks like there are dead cells with very intense PM-only TR signals. I suggest being more selective in the choice of images. If the bright dead cells are eliminated, the images could be contrasted more without saturation. All of the panels have the same pixel size; however, zooming in a bit more on the relevant individual cells would also make for clearer images. In this case, each individual panel would need its own scale bar.

We thank the reviewer for this suggestion. We have replaced the images.

5. Additional contrasting would benefit Figures 1b and 5d. They are quite difficult to see on my monitors.

We have revised figure 1b and 5d (now Fig 6d) in the revised manuscript.

6. The finding that Osh6 is the predominant supplier of PI4P is a bit surprising. Does over-expression of OSH7 produce a similar effect? or over-expression of OSH4. It would suggest that either Osh6 functions at a much higher rate or possibly that Sac1 more efficiently dephosphorylates PI4P transferred by Osh4. Perhaps the difference is between peripheral ER and nuclear ER. I don't believe this requires direct experimentation for this manuscript, but perhaps a sentence could be added to the text, raising the possibilities.

We agree with the reviewer that it is surprising that only Osh6 significantly contributes to PI4P exposure. While Osh7 is thought to act redundantly with Osh6 in PS transport from the ER to the plasma membrane, we found no evidence of its role in this pathway. However, overexpressing Osh7 in the osh6 Δ background or analyzing neo1-1 osh6 Δ osh7 Δ strains might reveal its contribution to PI4P flux. Additionally, PI4P produced at the Golgi by Pik1 and potentially transported to the ER by Osh4 does not appear to contribute to its extracellular exposure. We were concerned about drowning the readers with too many yeast genetic experiments and decided against doing a deeper analysis of Osh6/7 functional redundancy for this manuscript. We hope to return to this fascinating biology sometime soon.

Reviewer #2:

1. Line 87: The term “dnf” needs to be clearly described.

Thank you - we now provide a brief description of all the yeast flippases in the revised introduction section. (Page 2, line 61- 62) The term “dnf” stands for Drs2 Neo1 Family.

2. Lines 102-103: Please explain what “P4C” and “PH” stand for.

Page 3, lines 111-113. We now provide these definitions: "...we probed cells with recombinantly purified GFP-tagged P4C domain (PI4P binding domain from SidC), specifically binding to PI4P, and the PLC_{PH}, pleckstrin homology (PH) domain of phospholipase C delta, which binds to PI(4,5)P₂..."

3. Line 113: Why was only S221L selected for the assay among the subsets of mutants? Please explain the reasons.

All of the neomycin-sensitive *neo1* mutants grow as well as WT cells in the absence of neomycin, and the majority of the cells for each strain are exposing PI4P. Thus, it seemed highly unlikely that cell death accounted for PI4P exposure because we would have expected a growth defect if this were the case. Therefore, we chose *neo1-S221L* as representative of the subset for this experiment and found the expected result - the cells are viable and exclude propidium iodide. We didn't think it was necessary to repeat this for all the mutants. We often choose the Neo1-S221L variant because It is a little more specific for loss of PI4P asymmetry compared to the other neomycin sensitive variants. Neo1-S221L disrupts PI4P asymmetry with minimal impact on PE asymmetry and no measurable effect on PS asymmetry.

4. Lines 154-156: In Fig. 2h, the stimulation level by PS is similar to that of the combination of PS and PI4P in the 4th bar. Please provide an explanation for this phenomenon.

In our previous work, we demonstrated that PS and PE stimulate Neo1's ATPase activity in a dose-dependent manner¹. In this experiment, we tested the effects of adding PE or PI4P individually, both of which stimulate Neo1 ATPase activity. Based on older studies, the concentration of PE we used is close to saturation for stimulating ATPase activity. Thus, adding more substrate, either PE or PI4P, would not be expected to increase ATPase activity, which is what we observe. In contrast, if PI4P were an allosteric activator, it would convert "silent" Neo1 molecules in the population to an activated conformation and increase the apparent V_{max}. For example, Drs2 is autoinhibited by its C-terminal tail and PI4P relieves autoinhibition. For Drs2, the addition of substrate (PS) or PI4P alone provides minimal ATPase activation, while the combination of both lipids is required for full activation^{2,3}.

5. Fig 3: An overall structure of Neo1 is needed, and the RMSD of the structural superposition should be indicated. Additionally, the structure of Neo1 should be recolored to better distinguish it from Drs2.

We have updated Figure 4 to include the overall structure of Neo1-PI4P (New Fig. 4c). We have indicated the RMSD of the superposition (4.007 Å) in the legend, and have recolored Neo1 to differentiate it from Drs2 (New Fig. 4d). Thank you for these recommendations, they have improved the figure.

6. Lines 174-176: Are the residues coordinating the PI4P headgroup in Drs2 also conserved in Neo1? If so, will mutations of these residues impair the function, such as the PI4P-stimulated ATPase activity of Neo1? It is possible that PI4P functions both as a regulator and a substrate.

We thank the reviewer for this suggestion. We aligned the PI4P-binding region sequence of Neo1 with Drs2 and other P4-ATPases (Extended data Fig.7a). In Drs2, the glycerol backbone of PI4P is stabilized by interactions with positively charged residues, while selectivity for PI4P is mediated by Tyr1235 and His1236. Similarly, in Neo1, PI4P interacts with positively charged His472, His476, and Glu475, stabilizing the interaction. Substitution of these residues with alanine revealed that single mutants did not exhibit neomycin sensitivity. However, the double mutant (His472 and Glu475) displayed neomycin sensitivity comparable to Neo1-S221L mutants. These findings confirm the importance of these residues for PI4P transport (Fig. 4e). We also aligned the PI4P binding site residues in Drs2 with the Neo1 and these residues are not very well conserved in Neo1 (See alignment in below fig c, Extended Data Fig 7c).

Drs2 1219 RxxxWKxxxxxxxxxxYH 1236
 Neo1 1126 PxxxAKxxxxxxxxxxYA 1143

1210 BSB/MRB III
 465 21st Ave. South
 Nashville, TN 37232

tel: 615.322-2008
 fax 615.343-6707
<http://www.vanderbilt.edu/biosci/>

7. The subsets of mutants S211L, R247L, S452Q, and P456A should be mapped in the structure to better interpret how they affect the function of Neo1. Additionally, the PI4P-stimulated ATPase activity of these mutants needs to be assessed.

Thank you for these recommendations. We have updated the Neo1 structure figure to highlight the neomycin-sensitive mutant residues on the full structure (Fig. 4c) and their location along the substrate transport pathway in Figure 1b. Additionally, we tested the ATPase activity of Neo1 mutants with PI4P (new Figure 4b). Interestingly, the Neo1 mutants S221L and S452Q showed enhanced ATPase activity, while the R247L and P456A mutants reduced ATPase activity. Notably, the mutants with enhanced ATPase activity correspond to entry gate residues, whereas those with reduced ATPase activity are associated with exit gate residues. Please see the revised Discussion on page 10 for why we think the mutations are having this influence on PI4P-stimulated ATPase activity. Importantly, the entry gate in the Neo1 structures determined thus far is closed in the E2P conformation and is inaccessible to substrate. We suspect that binding interactions with Dop1 and Mon2, which are not present in the purified Neo1 samples, are needed to open the entry gate. The specific activity of substrate-stimulated ATPase activity for Neo1 *in vitro* is very low relative to other P4-ATPases, possibly because of the closed conformational state. Thus, we speculate that the entry gate mutations (S221L, S452Q) are opening up the entry gate, which is why we see enhanced PI4P-dependent ATPase activity. We are working to reconstitute the Neo1-Dop1-Mon2 complex for structural and biochemical studies but have not yet achieved this goal. We feel that the ability to measure substrate-stimulated ATPase activity with a fully-activated Neo1 will be critical to interpret the effects of the S221L and S452Q mutations. It will also be necessary to reconstitute substrate transport in proteoliposomes to measure potential competition between transport substrate. We have succeeded in reconstituting lipid transport with Drs2 but we have not been able to detect transport activity with purified Neo1 reconstituted in liposomes (again, we likely need the Dop1-Mon2-Neo1 complex). We hope to publish this fascinating biochemical observation here, but we feel the extensive analysis required to interpret this result accurately is beyond the scope of the current study (with its focus on the cellular function of Neo1/ATP9A on PI4P membrane asymmetry).

Reviewer #3

1. It is known that only some types of eukaryotic cells show neomycin sensitivity. Do those cell types display PI4P on the cell surface? Can the expression level of ATP9A/B explain the sensitivity of such cells? Is there any correlation between the level of ATP9A/B and sensitivity to neomycin among different cell types?

These are great questions. Neomycin is known to be both ototoxic and nephrotoxic^{4,5}. To investigate the correlation between neomycin sensitivity and ATP9A/9B expression, we analyzed ATP9A and ATP9B expression data in tissues from the GTEx Portal. The data suggest low ATP9A/B protein and RNA expression in kidney tissue, which are sensitive to neomycin. Similarly, RNA expression of ATP9A/B in commonly used cell lines show moderate expression in HeLa cells and low expression in HEK293 (kidney) cells (Extended data Fig.4c-e). We also confirmed ATP9A expression levels in HeLa and HEK293 cells through immunoblotting, which revealed high ATP9A expression in HeLa cells and low expression in HEK293 cells (Extended data 4b). To test the correlation between neomycin sensitivity and ATP9A expression, we assessed the neomycin sensitivity of HeLa cells, HEK293 cells, and ATP9A siRNA knockdown cells (New Figure 3). HeLa cells were resistant to neomycin up to 5000 $\mu\text{g}/\text{ml}$, but ATP9A knockdown in HeLa cells reduced neomycin resistance. In contrast, wild-type HEK293 cells were sensitive to neomycin, and ATP9A knockdown in HEK293 cells further increased sensitivity. We also examined PI4P exposure on the cell surface. Neomycin sensitivity in HeLa cells and ATP9A knockdown cells correlated with PI4P exposure. Wild-type HeLa cells did not expose PI4P on their surface, but ATP9A knockdown led to PI4P exposure, which contributed to neomycin sensitivity. Interestingly, while wild-type HEK293 cells were neomycin-sensitive, they did not exhibit PI4P exposure as measured with the GFP-SidC-P4C probe. However, ATP9A knockdown in HEK293 cells caused PI4P exposure on the outer membrane leaflet, further increasing neomycin sensitivity. Interestingly, the wild-type HEK293 cells were stained with Neo-TR, which did not stain HeLa cells. For both cell lines, ATP9a knockdown enhanced Neo-TR staining (New Figure 6f-i).

2. In Fig. 2i, biochemical analysis needs to be performed with different concentrations of substrates to correctly understand the enzyme and substrate relationship.

We agree and previously did this for PS and PE¹. We have now performed a dose response measurement with PI4P in the new Figure 4b.

3. In Fig. 2i, biochemical analysis needs to be done with a neomycin-sensitive mutant of NEO1 to clarify the enzyme and substrate relationship, especially for PI4P and PS or PE.

We tested the ATPase activity of Neo1 mutants with PI4P as suggested (Figure 4b)(Also see response to reviewer 2 point 7). Interestingly, the Neo1 mutants S221L and S452Q showed enhanced PI4P-stimulated ATPase activity, while the R247L and P456A mutants displayed a reduced ATPase activity. Notably, the mutants with enhanced ATPase activity correspond to entry gate residues, whereas those with reduced ATPase activity are associated with exit gate residues. This unexpected effect suggests that the entry gate mutations open this binding site, thereby increasing substrate-stimulated ATPase activity. In vivo, S221L and S452Q may disrupt PI4P transport by allowing PS or PE to outcompete PI4P at the entry gate binding site.

4. In Extended Data Fig. 2g, the data does not convincingly support the probe specificity. If the authors want to demonstrate probe specificity, it would be better to perform genetic manipulations to increase/decrease PI4P/PI(4,5)P₂ expression and show corresponding changes in signal intensity.

In this experiment, we aimed to demonstrate that GFP-SidC_{P4C} localizes to the Golgi and GFP-PLC_{PH} localizes to the plasma membrane in human cell lines. These probes are well-established in the literature for detecting PI4P/PI(4,5)P₂⁶⁻⁹ and the purpose here was to show the recombinant probes worked as expected. This was particularly important for the GFP-PLC-PH probe because the intact cells showed not signal. The control with fixed and permeabilized cells shows that the probe was working properly. Additionally, as shown in this manuscript in the extended data figure 3a, b, d, we used various yeast PI kinase mutants and the *sac1* mutant, which further confirm the specificity of these probes.

5. In Extended Data Figs. 2a, 2e, 2f, and 6b, the background signal varies significantly. Some data appears overly white, possibly due to excessive contrast manipulation. Can the authors clarify this point? In addition, what accounts for the differences between the left and right panels?

To clearly display the exposure of PI4P on the extracellular leaflets of the cells, we incubated the GFP-SidC_{P4C} and GFP-PLC_{PH} probe with the yeast spheroplast for the specific strain and captured the images. To visualize the images, we inverted the GFP channel using invert LUTs function in ImageJ. To be consistent with the analysis, very minimal adjustments were made in background and contrast. The right panel is the fluorescence signal intensity of the GFP-probe and left panel shows the DIC panel to display yeast cells. For clarity we have included this sentence in the figure legends.

6. In Extended Data Fig. 3, the legend description does not match the actual figure.

We thank the reviewer for catching the mistake. We have fixed this in the revised manuscript.

7. In Figs. 5b and 5c, it would be better to demonstrate localization to the Golgi and endosome by co-staining these organelles with markers.

To address the reviewer’s comment, we have now tested the localization of Neo-TR with a late Golgi marker Sec7-GFP and an endosomal marker GFP-Tlg1. We also quantified the colocalization of Neo-TR with Sec7 and Tlg1 (Extended data Fig. 9a,b).

8. In Fig. 5d, is latrunculin A treatment appropriate for endocytosis inhibition? Targeting actins affects functions other than endocytosis, so it might not necessarily prove the role of endocytosis. Is it possible to use other drugs such as Filipin, CPM, Dynasore, or Dyngo-5, or to perform genetic disruption of endocytosis machinery?

Latrunculin A (LatA) has been used in several studies to block endocytosis (Ayscough et al., 1997; Burston et al., 2009). It provides an excellent tool to study the temporal and rapid effects of endocytosis. We agree with the reviewer that targeting actin may impact other cellular functions. To address this, we knocked out the endocytic adaptor *SLA2* in the neomycin-sensitive mutant S221L and tested Neo-TR transport. The double mutant *neo1S221L sla2Δ* exhibited fewer intracellular structures and increased plasma membrane accumulation of Neo-TR compared to the *neo1S221L* mutant (Extended data fig. 9c,d). We thank the reviewers for this suggestion because these experiments helped us to test the neomycin sensitivity upon blocking endocytosis. Surprisingly, the *neo1S221L sla2Δ* strain still displayed sensitivity to neomycin, suggesting that blocking neomycin endocytosis does not protect cells from its toxicity (Extended Data Fig. 9e). It also raises the possibility that neomycin can directly cross the plasma membrane to intoxicate the cell.

9. In Fig. 5, can the Neo-TR assay be conducted in mammalian cells such as HeLa cells? This data would demonstrate that neomycin binding to PI4P is conserved in mammalian cells. Similarly to question 1, are neomycin-susceptible cells labeled with Neo-TR more?

We thank the reviewer for this excellent suggestion. We performed the Neo-TR assay in mammalian cell lines, including HeLa cells, HEK293 cells, and ATP9A knockdown cells. We observed that neomycin-resistant wild-type HeLa cells showed no binding of Neo-TR, while ATP9A knockdown cells, which expose PI4P, displayed significant binding of Neo-TR to the plasma membrane. Interestingly, both wild-type HEK293 cells and ATP9A knockdown HEK293 cells showed binding of Neo-TR to the plasma membrane and intracellular structures. Our data suggest that Neo-TR binds to neomycin-sensitive cells, and PI4P serves as a binding site for Neo-TR entry (Fig. 6f, g, h, i).

10. The role of Osh6 should be described in the manuscript. Why could Osh6Δ rescue the neo1-l phenotype?

We have clarified the role of Osh6 in the revised manuscript. Osh6 exchanges PI4P from the plasma membrane to the ER cytosol, swapping it with PS, which is then returned to the plasma membrane. Our data suggest that PI4P is sourced

from the plasma membrane, and Osh6 facilitates its exchange with PI4P/PS to the ER cytosolic leaflet. From there, it enters the ER lumen and progresses through the secretory pathway and exposes it to the extracellular leaflet of the cell. In *osh6Δ* cells, insufficient PI4P is supplied to the ER, leading to reduced PI4P exposure on the extracellular leaflet of the plasma membrane. This reduction causes *neo1-1osh6* mutants to be less sensitive to neomycin compared to *neo1-1*.

11. Can the authors show that PI4P exists on both the luminal and cytoplasmic sides of ER? This data supports the current hypothesis.

This is a great suggestion from the reviewer, and we are interested in probing PI4P in the ER lumen of the *neo1* mutant cells. We do not have a good way of performing this experiment. However, previous studies using the freeze-fracture method and electron microscopy have clearly shown luminal PI4P in *sac1* mutant cells¹⁰.

12. In Figs. 2i, 5c, and 5e, statistical comparison is lacking.

We have addressed this in the revised manuscript by including the statistical comparisons in Figures.

13. In Fig. 4a, the full names of each abbreviation need to be provided in the figure legend.

We have included the full names of each abbreviation in the figure legend for New Figure 5a in the revised manuscript.

Minor Points

1. In the abstract, it is better to mention earlier that PI4P serves as a neomycin receptor.
2. Line 42: references 2 and 5 should be superscripted.
3. Line 155: should refer to Fig. 2i, not 2h.
4. Lines 165 and 177: Fig. 3, there are no sub-panels.
5. Line 229: should refer to Fig. 5a since there is no other sentence referring to that panel.
6. In Extended Data Fig. 6a, more explanation is needed (what is neo1-2? what is 5-FOA?).
7. References: some names are capitalized (e.g., references 2 and 5).
8. In Extended Data Fig. 2, PI doesn't seem to be staining the nucleus.

We thank the reviewer for their attention to detail. We have included the textual or image changes as suggested in the revised manuscript.

References:

1. Bai, L. *et al.* Structural basis of the P4B ATPase lipid flippase activity. *Nat. Commun.* **12**, 5963 (2021).
2. Timcenko, M. *et al.* Structure and autoregulation of a P4-ATPase lipid flippase. *Nature* **571**, 366–370 (2019).

3. Bai, L. *et al.* Autoinhibition and activation mechanisms of the eukaryotic lipid flippase Drs2p-Cdc50p. *Nat. Commun.* **10**, 4142 (2019).
4. Kalbian, V. V. Deafness following oral use of neomycin. *South. Med. J.* **65**, 499–501 (1972).
5. Schibeci, A. & Schacht, J. Action of neomycin on the metabolism of polyphosphoinositides in the guinea pig kidney. *Biochem. Pharmacol.* **26**, 1769–1774 (1977).
6. Maib, H. *et al.* Recombinant biosensors for multiplex and super-resolution imaging of phosphoinositides. *J. Cell Biol.* **223**, (2024).
7. Luo, X. *et al.* Structure of the Legionella Virulence Factor, SidC Reveals a Unique PI(4)P-Specific Binding Domain Essential for Its Targeting to the Bacterial Phagosome. *PLOS Pathog.* **11**, e1004965 (2015).
8. Hammond, G. R. V, Ricci, M. M. C., Weckerly, C. C. & Wills, R. C. An update on genetically encoded lipid biosensors. *Mol. Biol. Cell* **33**, tp2 (2022).
9. Várnai, P. & Balla, T. Visualization of phosphoinositides that bind pleckstrin homology domains: calcium- and agonist-induced dynamic changes and relationship to myo-[3H]inositol-labeled phosphoinositide pools. *J. Cell Biol.* **143**, 501–510 (1998).
10. Muramoto, M. *et al.* Essential roles of phosphatidylinositol 4-phosphate phosphatases Sac1p and Sjl3p in yeast autophagosome formation. *Biochim. Biophys. Acta - Mol. Cell Biol. Lipids* **1867**, 159184 (2022).

Point-by-point response to Reviewers

Reviewer 1:

The authors have addressed by minor comments. The additional experiments to address the comments of Reviewers 2 and 3 only further strengthen an outstanding study. Congratulations on the elegant work.

We thank the reviewer for the positive assessment of our work.

Reviewer 2:

I have no further comments. I recommend this paper to be accepted.

We thank the reviewer for the positive assessment of our work.

Reviewer 3:

The authors have effectively addressed all the points I raised, making the article clearer and more compelling for a broad audience. I appreciate their efforts and congratulate them on these exciting findings. I now strongly support its publication in Nature Cell Biology.

We thank the reviewer for the positive assessment of our work.